# Adaptive 3D Reconstruction via Diffusion Priors and Forward Curvature-Matching Likelihood Updates

**Seunghyeok Shin**     **Dabin Kim**     **Hongki Lim**[*]
Department of Electrical and Computer Engineering, Inha University
{ssh8642, 1124db}@inha.edu, hklim@inha.ac.kr

## Abstract

Reconstructing high-quality point clouds from images remains challenging in computer vision. Existing generative models, particularly diffusion models, based approaches that directly learn the posterior may suffer from inflexibility—they require conditioning signals during training, support only a fixed number of input views, and need complete retraining for different measurements. Recent diffusion-based methods have attempted to address this by combining prior models with likelihood updates, but they rely on heuristic fixed step sizes for the likelihood update that lead to slow convergence and suboptimal reconstruction quality. We advance this line of approach by integrating our novel Forward Curvature-Matching (FCM) update method with diffusion sampling. Our method dynamically determines optimal step sizes using only forward automatic differentiation and finite-difference curvature estimates, enabling precise optimization of the likelihood update. This formulation enables high-fidelity reconstruction from both single-view and multi-view inputs, and supports various input modalities through simple operator substitution—all without retraining. Experiments on ShapeNet and CO3D datasets demonstrate that our method achieves superior reconstruction quality at matched or lower NFEs, yielding higher F-score and lower CD and EMD, validating its efficiency and adaptability for practical applications. Code is available at here.

## 1   Introduction

Three-dimensional reconstruction has become increasingly important across diverse applications including robotics, autonomous driving, augmented reality, and virtual environments. Among various 3D representations, point clouds serve as a fundamental data structure for representing objects and scenes due to their simplicity and flexibility. However, generating high-quality point clouds that accurately capture intricate details remains challenging, particularly when working with limited input information such as single-view images.

Recent advances in deep generative models, particularly diffusion models, have shown remarkable success in generating high-fidelity images [11, 14] and 3D data. Diffusion models use an iterative denoising process to progressively transform random noise into structured outputs, making them effective for capturing complex geometric patterns. In the domain of point cloud generation, researchers have begun exploring diffusion-based approaches with promising results [19, 39, 38, 24, 20, 21, 34].

While diffusion models offer powerful generative capabilities, applying them to 3D reconstruction presents unique challenges due to its nature as an inverse problem. In typical inverse problems (formulated as $\mathbf{y} = A\mathbf{x}$), iterative optimization methods solving least-squares objectives can determine optimal step sizes analytically using gradients that incorporate $A^\top$ (the adjoint of $A$). However, 3D object rendering represents a particularly challenging case where the rendering operator is complex and non-linear, making the computation of the adjoint operation intractable. Since classical step-size

---

[*]Corresponding author

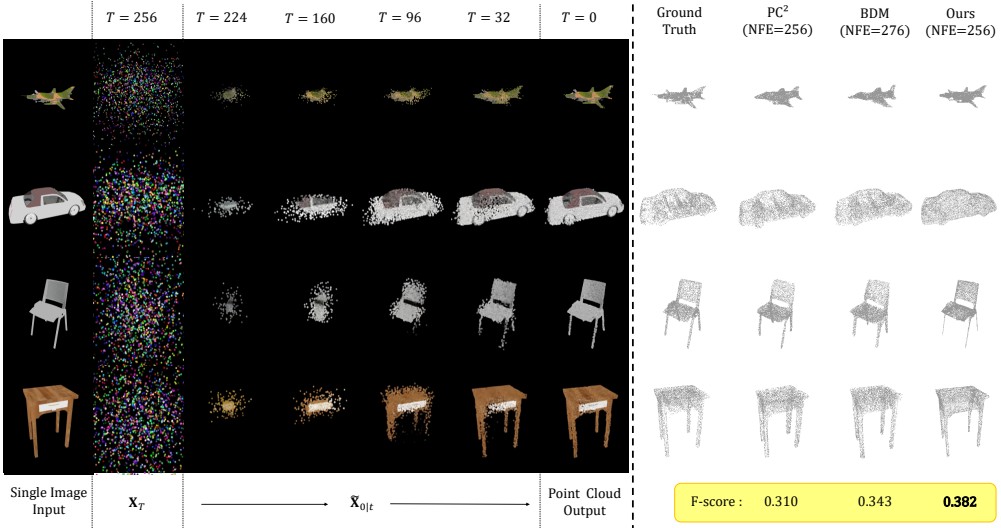

Figure 1: Left: Visualization of our diffusion process from random noise ($T = 256$) to final reconstructions ($T = 0$) for various object categories. Right: Comparison of point cloud reconstructions between Ground Truth, previous methods (PC² [20], BDM [34]), and our approach. Our method achieves higher fidelity reconstructions with better F-scores (0.382) than existing approaches while using fewer function evaluations, particularly excelling at preserving fine structural details.

formulas require an adjoint of the forward operator, the absence of a tractable renderer adjoint complicates step-size selection. This fundamental challenge impacts how researchers approach diffusion-based point cloud generation, especially when incorporating image-based guidance.

Current image-to-point-cloud methods predominantly learn the score of the posterior distribution $\nabla \log p(\mathbf{X}|\mathbf{y})$ directly, where $\mathbf{X}$ represents the point cloud and $\mathbf{y}$ represents image measurements. This direct approach incurs significant limitations: it necessitates including images as conditioning signals during training [20], restricts models to a fixed number of input views without specialized encoders [8], and requires computationally expensive retraining whenever measurement types change (e.g., from RGB images to depth maps).

A promising alternative approach [22] decomposes the posterior $p(\mathbf{X}|\mathbf{y})$ into a trainable prior $p(\mathbf{X})$ and an updatable likelihood $p(\mathbf{y}|\mathbf{X})$, employing Diffusion Posterior Sampling (DPS) [6] with gradient updates via $\nabla \log p(\mathbf{y}|\mathbf{X})$ for Gaussian splatting-based 3D reconstruction. While this decomposition is conceptually straightforward and modular, current implementations struggle with a critical limitation: determining appropriate step sizes for the likelihood updates. The non-linear nature of 3D rendering prevents analytical step size determination, leading existing methods to rely on heuristic, fixed step sizes [22]. This results in slow convergence, suboptimal reconstruction quality, and often necessitates additional 2D diffusion models for refinement, further complicating the pipeline.

To address these limitations, we present a novel approach that combines a point cloud diffusion model with Forward Curvature-Matching (FCM) optimization. Our approach, illustrated in Fig. 2, computes an adaptive step size using a Barzilai–Borwein rule and refines it with an Armijo backtracking condition, enabling more precise control. Our key insight is that by incorporating FCM's principled, curvature informed step size determination into the diffusion sampling process without any adjoint operations, we can effectively navigate the complex optimization landscape of 3D reconstruction.

Unlike previous DPS-based methods that rely on heuristic step sizes for the likelihood update, our approach employs FCM optimization to dynamically determine optimal step sizes. The key innovation is our reliance solely on the differentiable forward pass for curvature-informed step-size determination, obviating the adjoint. This enhancement enables significantly more efficient and accurate optimization during the diffusion sampling process. The technical contributions of our work include:

- We integrate the FCM method with the reverse process of diffusion models, enabling high-fidelity point cloud reconstruction that accurately matches input images.
- Our gradient-based updates are not constrained by the number of input images, allowing for point cloud reconstruction from either single-view or multi-view images without modifying the base model.
- Our method can be applied to various measurement modalities (such as RGB image to 3D object or depth map to 3D object) by simply substituting the appropriate operator rather than retraining the entire model, significantly enhancing flexibility and efficiency.

We demonstrate the effectiveness of our approach by reconstructing colored point clouds from both synthetic and real-world datasets. Our method achieves more accurate reconstruction with fewer neural function evaluations (NFEs) compared to existing techniques, validating the efficiency of our FCM-based likelihood optimization. We further demonstrate the adaptability of our approach by applying it to both multi-view reconstruction and depth map to point cloud generation without retraining, highlighting its potential for diverse applications. The remainder of this paper is organized as follows: Section 2 reviews related work, Section 3 presents the proposed method, Section 4 details our experimental results, and Section 5 concludes with a discussion of future directions.

## 2 Related Work

**3D Reconstruction from Images.** As interest in 3D content creation continues to grow, research on reconstructing 3D shapes from 2D observations has advanced significantly. This challenging task requires inferring complete 3D structures, including both visible and occluded regions, from limited viewpoints. The difficulty is compounded by the scarcity of large-scale 3D datasets.

Various 3D representations have been explored for reconstruction, each with distinct advantages: mesh-based methods [13, 32, 3] offer compact representation but struggle with topological complexity; voxel-based approaches [15] provide a regular structure but face resolution limitations; point cloud methods [20, 34, 16] offer flexibility with additional rendering requirements; implicit functions [23, 10, 5, 12] enable high-quality rendering but are computationally intensive; and Gaussian splatting techniques [31, 30, 22] balance quality and efficiency.

Point cloud generative models have evolved from early GAN-based [1, 9, 28] and VAE-based [36] approaches to more recent diffusion-based methods. Diffusion models offer several advantages: stable training dynamics, high-quality generation capabilities, flexibility in conditioning, and a strong probabilistic foundation. The seminal work by Luo et al. [19] introduced diffusion models for point cloud generation, with subsequent research extending these methods for various applications [38, 17].

Building on these advancements in 3D generative modeling, recent diffusion-based approaches have significantly advanced image-to-point cloud reconstruction. PC$^2$ [20] performs single-view reconstruction by denoising a point cloud with projection conditioning, which ensures geometric consistency between the reconstruction and input view. However, it directly learns the posterior distribution, requiring images during training and limiting adaptability to varying input conditions. Bayesian Diffusion Models (BDM) [34] offer a complementary perspective by factorizing the 3D reconstruction task into a learned score of the prior $\nabla \log p(\mathbf{X})$ trained solely on 3D shapes and a learned score of the posterior $\nabla \log p(\mathbf{X}|\mathbf{y})$ trained with paired image–shape data. During inference, the prior and posterior models exchange intermediate outputs over multiple denoising steps. While this "fusion-with-diffusion" paradigm is effective, BDM relies on a PC$^2$-like trained posterior score function that requires images during training, thus limiting its adaptability to varying input modalities.

**Diffusion Posterior Sampling.** Diffusion Posterior Sampling (DPS) [6] proposes a framework for solving inverse problems using diffusion models without retraining for each new measurement type. This approach decomposes the posterior $p(\mathbf{X}|\mathbf{y})$ into a pre-trained prior $p(\mathbf{X})$ and an adaptable likelihood term $p(\mathbf{y}|\mathbf{X})$. During sampling, the intermediate predictions are adjusted using gradient updates from the likelihood term.

Recent works applying DPS to 3D reconstruction include GSD [22], which uses DPS with Gaussian Splatting for view-guided 3D generation. However, these methods rely on heuristic, manually-tuned step sizes for the likelihood update, which often requires careful calibration for each task and can lead to suboptimal convergence or reconstruction quality.

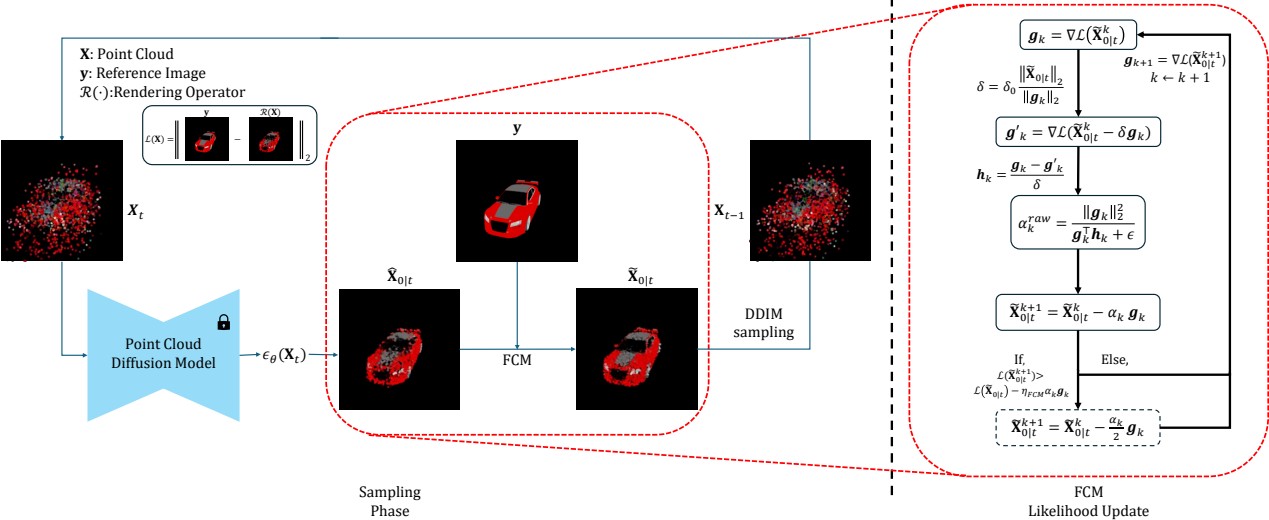

Figure 2: Overview of our FCM-guided point cloud diffusion framework. The sampling phase (left) shows how the diffusion model progressively transforms random noise $\mathbf{X}_t$ into structured point clouds through DDIM sampling. The FCM likelihood update (right) illustrates our key innovation— dynamically determining optimal step sizes for the likelihood gradient $\nabla\|\mathbf{y} - R(\hat{\mathbf{X}}_{0|t})\|_2$. This principled optimization approach enables high-fidelity reconstruction that accurately matches input images while requiring fewer function evaluations than existing methods.

**Adaptive Step Size Methods in Optimization.**   Our FCM approach has roots in several foundational optimization techniques while introducing novel algorithmic elements. In numerical optimization, determining appropriate step sizes is a well-studied challenge with various classical solutions. Quasi-Newton methods [25] approximate the Hessian using rank-one or rank-two updates (e.g., BFGS, L-BFGS [18]), but require matrix storage and operations. Barzilai-Borwein (BB) methods [4] provide scalar approximations to the secant equation using the differences of consecutive iterates and gradients. Line search techniques with Armijo [2] or Wolfe conditions [33] ensure sufficient descent but typically involve multiple function evaluations.

Building on these foundations, FCM introduces several innovations specifically for diffusion-based 3D reconstruction: (1) a scale-adaptive curvature probe ($\delta_k = \delta_0 \cdot \frac{\|\mathbf{x}_k\|}{\|\mathbf{g}_k\|}$) that automatically calibrates to the geometry of point clouds and gradient magnitudes, (2) a forward-difference directional curvature estimate that requires no adjoint operations of the renderer—critical for complex neural renderers where adjoint computation is intractable, (3) a robust BB-inspired step-size computation combined with principled capping that offers theoretical guarantees, and (4) a "once-only" Armijo check.

## 3   Method

Our goal is to perform high-quality, flexible 3D reconstruction by decomposing the posterior distribution $p(\mathbf{X} \mid \mathbf{y})$ into a learned prior $p_\theta(\mathbf{X})$ and a likelihood update $p(\mathbf{y} \mid \mathbf{X})$ that does not require separate training. We train only the score of the prior $\nabla \log p_\theta(\mathbf{X})$ on unlabeled 3D data. Then, at inference, we incorporate the measurement information (e.g., single-view or multi-view images, depth maps) through an adaptive Forward Curvature-Matching (FCM) update, which approximates $\nabla \log p(\mathbf{y} \mid \mathbf{X})$.

Any forward operator $\mathcal{R}$ (e.g., a differentiable renderer for images or a map from 3D to depth measurements) can be plugged in to guide the generation of point clouds via the same trained diffusion prior. This design separates the learned model from the measurement modality, eliminating the need for retraining whenever the measurement operator changes. In this section, we detail our method in four parts. First, we describe how we train the diffusion model $\nabla \log p_\theta(\mathbf{X})$. Next, we present our differentiable renderer $\mathcal{R}$ for the image-based scenario. We then introduce the FCM-based likelihood update, highlighting why FCM is needed in non-linear settings and how step sizes are optimally determined through a principled approach. Finally, we extend the method to the multi-view setting.

## 3.1 Diffusion Prior for Point Clouds

We begin by training a diffusion model $p_\theta(\mathbf{X})$ on a large dataset of colored point clouds. Following the standard DDPM [11] framework, we define a forward diffusion process that corrupts a clean point cloud $\mathbf{X}_0$ into $\mathbf{X}_T$ with Gaussian noise over $T$ timesteps. The reverse process is modeled by a neural network that estimates the noise at each timestep. Formally, in the forward process:

$$q(\mathbf{X}_t \mid \mathbf{X}_{t-1}) = \mathcal{N}\big(\mathbf{X}_t; \sqrt{1 - \beta_t}\,\mathbf{X}_{t-1}, \beta_t\,\mathbf{I}\big), \tag{1}$$

where $\beta_t$ is a variance schedule. This process can be written in closed form from $\mathbf{X}_0$:

$$q(\mathbf{X}_t \mid \mathbf{X}_0) = \mathcal{N}\big(\mathbf{X}_t; \sqrt{\bar{\alpha}_t}\,\mathbf{X}_0,\, (1 - \bar{\alpha}_t)\,\mathbf{I}\big), \tag{2}$$

with $\bar{\alpha}_t = \prod_{s=1}^{t}(1 - \beta_s)$. The reverse process approximates $p_\theta(\mathbf{X}_{t-1} \mid \mathbf{X}_t)$ via a learned Gaussian:

$$p_\theta(\mathbf{X}_{t-1} \mid \mathbf{X}_t) = \mathcal{N}\big(\mathbf{X}_{t-1};\, \mu_\theta(\mathbf{X}_t, t),\, \Sigma_\theta(\mathbf{X}_t, t)\big). \tag{3}$$

During training, we minimize the simplified loss:

$$L = \mathbb{E}_{t, \mathbf{X}_0, \boldsymbol{\epsilon}}\Big[\|\boldsymbol{\epsilon} - \boldsymbol{\epsilon}_\theta(\mathbf{X}_t, t)\|^2\Big], \tag{4}$$

where $\boldsymbol{\epsilon} \sim \mathcal{N}(\mathbf{0}, \mathbf{I})$.

Once trained, we use the DDIM sampler [29] for inference, generating point clouds from noise in fewer steps. Let $\epsilon_\theta^{(t)}(\mathbf{X}_t)$ be the noise estimate at step $t$. Then the DDIM update from $\mathbf{X}_t$ to $\mathbf{X}_{t-1}$ is:

$$\mathbf{X}_{t-1} = \sqrt{\bar{\alpha}_{t-1}}\,\hat{\mathbf{X}}_{0|t} + \sqrt{1 - \bar{\alpha}_{t-1} - \sigma_t(\eta)^2}\,\epsilon_\theta^{(t)}(\mathbf{X}_t) + \sigma_t(\eta)\,\boldsymbol{\epsilon}_t, \tag{5}$$

where

$$\hat{\mathbf{X}}_{0|t} = \frac{\mathbf{X}_t - \sqrt{1 - \bar{\alpha}_t}\,\epsilon_\theta^{(t)}(\mathbf{X}_t)}{\sqrt{\bar{\alpha}_t}}$$

and

$$\sigma_t(\eta) = \eta\sqrt{\frac{(1 - \bar{\alpha}_{t-1})}{(1 - \bar{\alpha}_t)}}\,\sqrt{1 - \frac{\bar{\alpha}_t}{\bar{\alpha}_{t-1}}},$$

is a variance term controlling the sampling stochasticity. This DDIM sampler, combined with our trained model, provides a 3D prior that can generate plausible point clouds.

## 3.2 Differentiable Renderer as the Measurement Operator

Our method only requires that $\mathcal{R}$ be differentiable, so both $\mathcal{R}(\mathbf{X})$ and its gradient $\nabla_\mathbf{X}\|\mathbf{y} - \mathcal{R}(\mathbf{X})\|_2$ can be computed. In this section we introduce a forward operator $\mathcal{R}$ that projects a point cloud $\mathbf{X}$ into 2D measurements.

A point cloud $\mathbf{X}$ comprises points $\{(x_i, y_i, z_i, \mathbf{f}_i)\}$, where $(x_i, y_i, z_i)$ are 3D coordinates and $\mathbf{f}_i$ includes attributes such as color. Each point is projected onto the 2D image plane using known camera parameters. At each pixel $(u, v)$, $\mathcal{R}$ collects the $K$ points with the smallest depth values $z_i$ (i.e., the nearest points along the viewing direction) and blends their colors via alpha compositing:

$$\mathcal{R}_{\text{color}}(\mathbf{X})[u, v] = \sum_{i=1}^{K}\Big(\alpha_i \prod_{j=1}^{i-1}(1 - \alpha_j)\Big)\mathbf{f}_i. \tag{6}$$

Here, the opacity $\alpha_i$ is computed from the image space footprint as

$$\alpha_i = 1 - \frac{\rho_i^2}{r^2}, \tag{7}$$

where $r$ is the radius of the rasterizer and $\rho_i$ is the Euclidean distance between the center of the pixel and the projected position of the point in the image space. The product term $\prod_{j=1}^{i-1}(1 - \alpha_j)$ ensures that closer points dominate the final color, while partially occluded points contribute less. Repeating this calculation for each pixel $(u, v)$ yields a 2D image matching the resolution of $\mathbf{y}$.

In addition to color-based rendering, $\mathcal{R}$ can produce a depth map by applying inverse-square weighting to each point's distance. At each pixel $(u, v)$, the depth is computed from the same set of $K$ nearest points:

$$\mathcal{R}_{\text{depth}}(\mathbf{X})[u, v] = \frac{\sum_{i=1}^{K}\frac{1}{d_i}}{\sum_{i=1}^{K}\frac{1}{d_i^2}}, \tag{8}$$

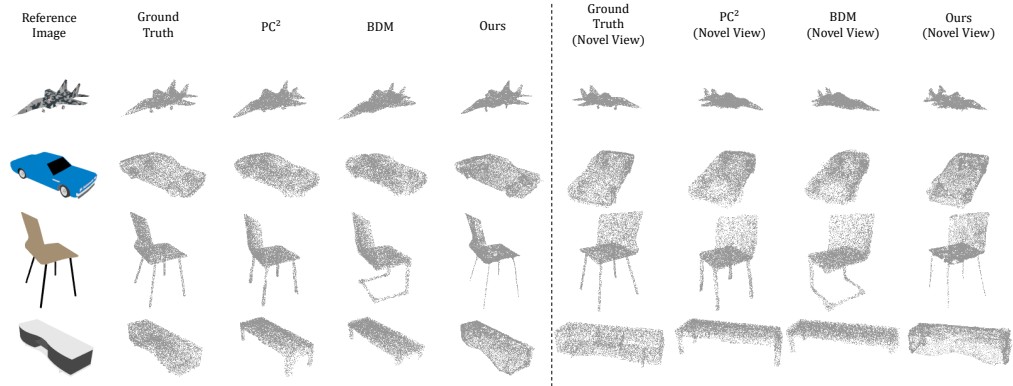

| | Reference Image | Ground Truth | PC² | BDM | Ours | Ground Truth (Novel View) | PC² (Novel View) | BDM (Novel View) | Ours (Novel View) |

Figure 3: Qualitative comparison of single-view 3D reconstructions on the ShapeNet dataset. The figure displays point cloud reconstructions from our method, PC², and BDM for various object categories, highlighting the superior detail and accuracy of our approach.

| Category | EMD($\times 10$) | | | CD($\times 10$) | | | F-score | | |
|---|---|---|---|---|---|---|---|---|---|
| | PC² [20] | BDM [34] | Ours | PC² [20] | BDM [34] | Ours | PC² [20] | BDM [34] | Ours |
| airplane | 0.587 | 0.577 | **0.476** | 0.399 | 0.417 | **0.378** | 0.498 | **0.543** | 0.543 |
| car | 0.565 | 0.723 | **0.517** | 0.558 | 0.664 | **0.460** | 0.262 | 0.289 | **0.386** |
| chair | 0.701 | **0.643** | 0.662 | 0.636 | **0.613** | 0.679 | 0.241 | 0.271 | **0.282** |
| table | 0.735 | **0.647** | 0.691 | 0.703 | **0.656** | 0.727 | 0.240 | 0.268 | **0.319** |
| Average | 0.647 | 0.648 | **0.587** | 0.574 | 0.588 | **0.561** | 0.310 | 0.343 | **0.382** |

Table 1: Quantitative evaluation of single-view 3D reconstruction on the ShapeNet dataset. NFEs were matched equally across our method, PC², and reconstruction model of BDM ($T = 256$). For BDM, additional NFEs were incurred due to the prior model ($T = 20$).

so that points closer to the camera have a larger influence on the final depth. Repeating this process for each pixel yields a 2D depth map matching the resolution of $\mathbf{y}$.

Because we do not learn a dedicated score function for $\nabla \log p(\mathbf{y} \mid \mathbf{X})$, different operators $\mathcal{R}$ can be swapped in with minimal effort. If $\mathbf{y}$ is a single-view image, then $\mathcal{R} = \mathcal{R}_{\text{color}}$ with a single camera. For multi-view input, each view is rendered separately and their pixel or feature errors are averaged, as described in Section 3.4. If $\mathbf{y}$ is a depth map, then $\mathcal{R} = \mathcal{R}_{\text{depth}}$ from Eq. (8).

### 3.3 Likelihood Update via Forward Curvature-Matching

In standard diffusion posterior sampling (DPS) [6], one iteratively updates the current sample $\mathbf{X}_t$ with a term proportional to the gradient $\nabla_{\mathbf{X}} \log p(\mathbf{y} \mid \mathbf{X})$. However, for complex, non-linear forward operators $\mathcal{R}$, determining an appropriate step size is non-trivial. Previous approaches resort to heuristics [6] or empirically tuned factors [22] to balance the data fidelity term with the learned diffusion prior. While this can be effective, it may hamper convergence speed or degrade reconstruction quality if not carefully tuned.

To address these limitations, we propose Forward Curvature-Matching (FCM), a novel algorithm designed specifically for diffusion-based 3D reconstruction. The development of FCM was guided by key requirements: working without adjoint operations (intractable for neural renderers), maintaining predictable computational cost, and using universal parameters across different reconstruction tasks.

Our approach relies on a key insight: we can estimate curvature information through a scaled directional probe without requiring full Hessian approximations [25]. For the measurement loss $\mathcal{L}(\mathbf{x}) = \|\mathbf{y} - \mathcal{R}(\mathbf{x})\|_2$, given the current estimate $\mathbf{x}_k$ and gradient $\mathbf{g}_k = \nabla \mathcal{L}(\mathbf{x}_k)$, we compute:

$$\delta_k = \delta_0 \cdot \frac{\|\mathbf{x}_k\|}{\|\mathbf{g}_k\|}, \qquad \mathbf{x}'_k = \mathbf{x}_k - \delta_k \cdot \mathbf{g}_k, \tag{9}$$

$$\mathbf{g}'_k = \nabla \mathcal{L}(\mathbf{x}'_k), \qquad \mathbf{h}_k = \frac{\mathbf{g}_k - \mathbf{g}'_k}{\delta_k} \tag{10}$$

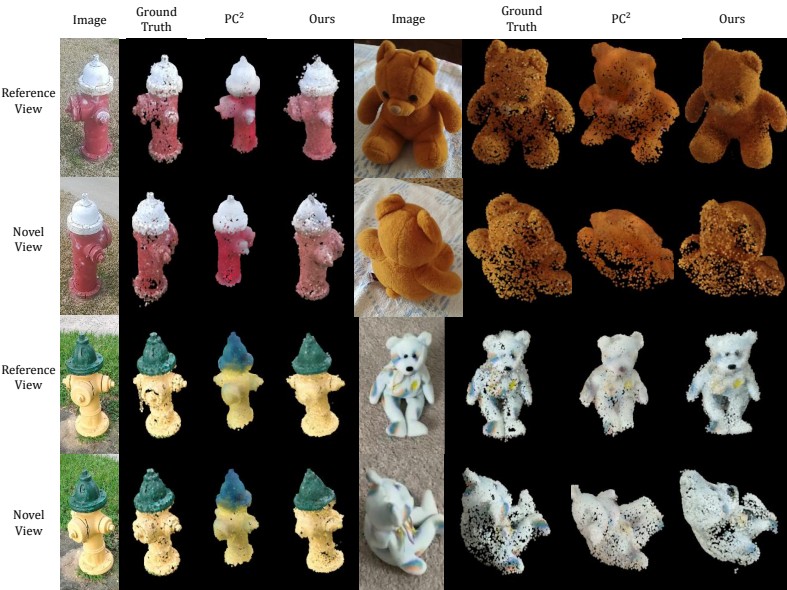

Figure 4: Comparison of rendered images from reconstructed point clouds on the CO3D dataset. The figure shows renderings from our method and PC$^2$, illustrating the higher fidelity and better preservation of details in our reconstructions.

This $\mathbf{h}_k$ approximates $\nabla^2 \mathcal{L}(\mathbf{x}_k) \cdot \mathbf{g}_k$ along the gradient direction. The scale-adaptive probe ($\delta_k$) automatically calibrates to the geometry of the point cloud, a crucial advantage over traditional finite-difference approaches [7].

We then compute a Barzilai–Borwein-inspired [4] step size, modified for robustness:

$$\alpha_k^{\text{raw}} = \frac{\|\mathbf{g}_k\|^2}{\langle \mathbf{g}_k, \mathbf{h}_k \rangle + \varepsilon}, \quad \alpha_k = \min\{\alpha_k^{\text{raw}}, 1/L\} \tag{11}$$

where $\varepsilon = 10^{-12}$ and $L$ is the Lipschitz constant of $\nabla \mathcal{L}$. The capping mechanism ensures stability while maintaining theoretical guarantees. Unlike classical line searches that require multiple function evaluations [25], we incorporate a single Armijo check [2]: if $\mathcal{L}(\mathbf{x}_k - \alpha_k \mathbf{g}_k) > \mathcal{L}(\mathbf{x}_k) - \eta_{\text{FCM}} \cdot \alpha_k \cdot \|\mathbf{g}_k\|^2$, we halve $\alpha_k$ once and accept.

This design yields a fixed computational cost of exactly two backward and three forward passes per step—significantly more efficient than traditional optimization methods like L-BFGS [18] or Wolfe line searches [33] with unpredictable evaluation counts.

### 3.3.1 Theoretical Guarantees

Our approach is built on the following assumptions, which are typically satisfied in the context of 3D reconstruction:

**Assumption 3.1** (Smoothness). *$\mathcal{L}$ is $L$-smooth:* $\|\nabla\mathcal{L}(\mathbf{u}) - \nabla\mathcal{L}(\mathbf{v})\| \leq L \cdot \|\mathbf{u} - \mathbf{v}\|$.

**Assumption 3.2** (Lower bound). *$\mathcal{L}_{\inf} := \inf_{\mathbf{x}} \mathcal{L}(\mathbf{x}) > -\infty$.*

**Assumption 3.3** (Local convexity). *$\mathcal{L}$ is convex on the set of iterates (which is typically small or "benign" in practice).*

This approach provides theoretical guarantees on convergence and optimality, as captured in the following theorem:

**Theorem 3.4** (Guaranteed Loss Decrease). *Let $c = \min\{\frac{\eta_{FCM}}{2L}, \frac{1}{8L}\}$. Our FCM algorithm ensures:*

$$\mathcal{L}(\mathbf{x}_{k+1}) \leq \mathcal{L}(\mathbf{x}_k) - c \cdot \|\nabla\mathcal{L}(\mathbf{x}_k)\|^2 \tag{12}$$

When integrated into the DDIM sampling process, FCM preserves the contraction properties of diffusion models:

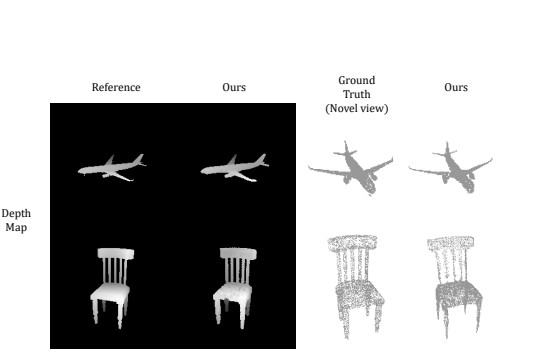

Figure 5: Reconstruction from depth maps. The figure showcases point cloud reconstructions generated from depth map inputs.

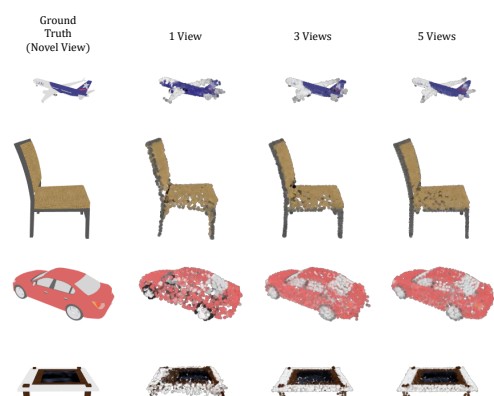

Figure 6: Qualitative results of multi-view reconstruction. The figure displays point cloud reconstructions using varying numbers of input views, demonstrating the enhancement in reconstruction quality as more views are incorporated.

**Proposition 3.5** (Contraction Preservation). *Under Assumptions 3.1–3.3 and $\alpha_k \leq 1/L$, the combined DDIM+FCM operator remains a contraction in expectation, thus preserving the diffusion contraction property.*

Our FCM method uses fixed constant $\eta_{\text{FCM}} = 10^{-4}$ for all tasks, this principled approach leads to faster convergence and higher-quality reconstructions compared to methods that rely on heuristic step sizes. Detailed proofs and additional theoretical analysis are provided in the Appendix.

### 3.4 Multi-View Reconstruction

The same FCM-based likelihood update extends naturally to multi-view reconstruction. Suppose we have $N$ images $\{\mathbf{y}_i\}_{i=1}^{N}$ with known camera parameters. We define

$$\mathcal{L}_{\text{MV}}(\mathbf{X}) = \frac{1}{N} \sum_{i=1}^{N} \left\| \mathbf{y}_i - \mathcal{R}_i(\mathbf{X}) \right\|_2, \tag{13}$$

where $\mathcal{R}_i$ is the differentiable renderer for the $i$-th viewpoint. The gradient $\nabla_{\mathbf{X}} \mathcal{L}_{\text{MV}}(\mathbf{X})$ can be used in Algorithm 1 (replacing the single-view line $\|\mathbf{y} - \mathcal{R}(\cdot)\|$ with the multi-view average). As the number of views grows, reconstruction quality improves, yet the diffusion prior remains the same, illustrating the modality-agnostic nature of our approach.

By training only the diffusion prior on unlabeled 3D shapes and introducing an FCM-based likelihood update with an arbitrary forward operator $\mathcal{R}$, we achieve a flexible, adaptive 3D reconstruction pipeline. The FCM approach ensures stable and fast convergence even with non-linear rendering operators, outperforming fixed-step DPS approaches.

## 4 Experiments

We evaluate the reconstructed point clouds using three different metrics: Earth Mover's Distance (EMD), L-1 Chamfer Distance (CD), and F-score at a threshold of 0.01. Details of the implementation are provided in the appendix.

**ShapeNet.** In our method, colors are essential during the rendering process. However, sampling colored point clouds from mesh-based objects is a challenging task. To address this, we train our model using the dataset provided by KeypointNet [37]. The color information in the KeypointNet point cloud does not correspond to the actual mesh color in ShapeNet. Instead, the model assigns colors according to object parts.

We perform our evaluation using the categories {*airplane, car, chair, table*} from the ShapeNet rendered image dataset [35].

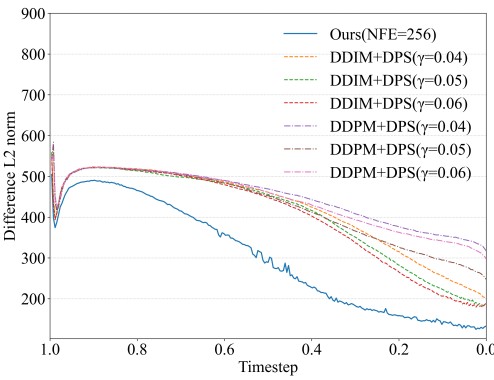

| Views | EMD($\times 10$) | CD($\times 10$) | F-score |
|---|---|---|---|
| 1 | 0.587 | 0.561 | 0.382 |
| 3 | 0.436 | 0.386 | 0.512 |
| 5 | **0.425** | **0.361** | **0.548** |

Table 2: Impact of the number of input views on reconstruction performance. The table presents scores for reconstructions using 1, 3, and 5 views, showing improved quality with additional views.

| Method | EMD($\times 10$) | CD($\times 10$) | F-score |
|---|---|---|---|
| DDPM + DPS | 0.674 | 0.688 | 0.337 |
| DDIM + DPS | 0.716 | 0.728 | 0.312 |
| Ours | **0.587** | **0.561** | **0.382** |

Figure 7: Convergence analysis during sampling. $\gamma$ is step size of DPS-update. The plot shows the L2 norm difference between the reference image and the rendered image ($\|\boldsymbol{y} - \mathcal{R}(\hat{\boldsymbol{X}}_{0|t})\|_2$) over diffusion timesteps for our method and other sampling approaches, illustrating more stable convergence of our FCM-based method.

Table 3: Comparison with DPS-based methods. The table presents reconstruction metrics for our method versus DDPM+DPS and DDIM+DPS, demonstrating our approach's superior performance with fewer NFEs.

**CO3D.** The CO3D dataset is a large-scale collection of real-world multi-view images from common object categories. It provides a colored point cloud obtained using COLMAP from multi-view images, which is then used for model training and evaluation. We perform our evaluation using the categories *hydrant* and *teddybear* from the CO3D dataset.

### 4.1 Quantitative Results

We evaluate the performance of reconstruction in the ShapeNet dataset. In Tab. 1, our method is compared with PC$^2$ [20] and BDM [34]. In the original paper, BDM is evaluated using 4,096 points, whereas PC$^2$ is evaluated using 8,192 points. In this work, we adopt the evaluation approach of PC$^2$ for quantitative experiments. For BDM, we adopted the blending method that achieved the best results in their study and used PC$^2$ as the reconstruction model. Our method achieves the best results in all metrics. In this experiment, we ensured that the NFEs for all other models were set similarly for a fair comparison. Detailed comparisons with the settings proposed by their studies and quantitative results on CO3D are provided in the appendix.

### 4.2 Qualitative Results

In Fig. 3 we show the reconstructed point clouds of different models using the ShapeNet dataset. In Fig. 4 we present a comparison of the rendered results of reconstructed colored point clouds using the CO3D dataset. Other models fail to accurately follow the given image in their rendering results for the reference view, instead focusing on generating a plausible object within the learned category. However, our method achieves the highest level of detail for the reference image.

### 4.3 Adaptivity Analysis

Our method has the advantage of performing various tasks without requiring retraining of the model. In this section, we demonstrate this capability through multi-view reconstruction and depth map reconstruction. The models used in this section are the same as those used in the previous section for the ShapeNet dataset. Fig. 6 and Tab. 2 illustrate the effectiveness of our method in multi-view reconstruction. As the number of views increases, the generated point cloud becomes more refined, demonstrating the improved quality of reconstruction. Fig. 5 presents the results of applying our method to depth maps rendered using Eq. 8. The results show high fidelity to the reference depth map and the ability to generate natural-looking objects.

### 4.4 Ablation Study

To show the effectiveness of our method, we compare with other DPS-based methods. Fig. 7 and Tab. 3 compare our method with DPS-based approaches. Fig. 7 presents the plot of the difference

in L2 norm between the reference image and the rendered image during the sampling process over timesteps. We observed that both DDPM+DPS and DDIM+DPS methods achieve their best performance at the step size of 0.05. The reason DPS-based methods struggle to follow the reference image is that they update with a fixed step size, leading to suboptimal convergence. It demonstrates that our method converges more optimally compared to other approaches. As shown in Tab. 3, our method achieves the best point cloud reconstruction performance. Since the DDPM sampling process does not approximate $\hat{\mathbf{X}}_0$, and the iterative FCM updates from noisy $\mathbf{X}_t$ using measurement $\mathbf{y}$ are not ideal, we exclude the DDPM+FCM scheme from our comparison. Qualitative comparisons with DPS-based methods are provided in the appendix.

## 5 Conclusion

In this paper, we proposed the novel point cloud diffusion sampling approach for adaptive 3D reconstruction. Our method reconstructs the colored point cloud by updating it using likelihood $\nabla \log p(\mathbf{y} \mid \mathbf{X})$ with given images through FCM during the reverse process of the point cloud diffusion model. In our experiments, we qualitatively demonstrate high-fidelity reconstruction of reference images with color, generating high-quality point cloud structures compared to prior works. Moreover, we quantitatively surpass previous works in point cloud reconstruction performance. Our method is applicable to various tasks, demonstrating its versatility. Additionally, it can be extended to different domains (e.g., Gaussian Splatting, meshes, etc.), highlighting its adaptability. As future work, we are interested in exploring larger datasets across diverse domains.

## Acknowledgement

This work was supported in part by the National Research Foundation of Korea (NRF) grant funded by the Korea government (MSIT) (RS-2025-24683103), in part by Korea Basic Science Institute (National research Facilities and Equipment Center) grant funded by the Ministry of Science and ICT (No. RS-2024-00401899), in part by Institute of Information & communications Technology Planning & Evaluation (IITP) under the Leading Generative AI Human Resources Development (IITP-2025-RS-2024-00360227) grant funded by the Korea government (MSIT), and in part by Institute of Information & communications Technology Planning & Evaluation (IITP) grant funded by the Korea government (MSIT) (No.RS-2022-00155915, Artificial Intelligence Convergence Innovation Human Resources Development (Inha University)).

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

## A.1 Algorithm

---

**Algorithm 1** DDIM Sampling with FCM Likelihood Update

---

**Require:** Trained noise predictor $\boldsymbol{\epsilon}_{\theta^\star}$, Measurement $\mathbf{y}$, Total diffusion steps $T$, DDIM cumulative schedule $\{\bar{\alpha}_t\}_{t=1}^T$, Stochasticity coefficient $\eta$, Differentiable renderer $\mathcal{R}$, FCM hyper-parameters $\delta_0$, $\eta_{\mathrm{FCM}}$, Lipschitz bound $L$, Numerical stabilizer $\varepsilon$

1: $\boldsymbol{X}_T \sim \mathcal{N}(\mathbf{0}, \mathbf{I})$         ▷ initialize with noise
2: **for** $t = T$ **downto** 1 **do**
    **(a) DDIM prior prediction**
3:     $\hat{\boldsymbol{\epsilon}}_t \leftarrow \boldsymbol{\epsilon}_{\theta^\star}(\boldsymbol{X}_t, t)$
4:     $\hat{\boldsymbol{X}}_{0|t} \leftarrow (\boldsymbol{X}_t - \sqrt{1 - \bar{\alpha}_t}\, \hat{\boldsymbol{\epsilon}}_t)/\sqrt{\bar{\alpha}_t}$
    **(b) FCM likelihood refinement**
5:     $\mathbf{x}_0 \leftarrow \hat{\boldsymbol{X}}_{0|t}$
6:     **for** $k = 0$ **to** $K - 1$ **do**     ▷ $K$=4 outer refinements
7:         $\mathbf{g}_k \leftarrow \nabla_{\mathbf{x}_k} \left\| \mathbf{y} - \mathcal{R}(\mathbf{x}_k) \right\|_2$
8:         $\delta_k \leftarrow \delta_0 \left\| \mathbf{x}_k \right\| / \left\| \mathbf{g}_k \right\|$
9:         $\mathbf{x}'_k \leftarrow \mathbf{x}_k - \delta_k\, \mathbf{g}_k$
10:        $\mathbf{g}'_k \leftarrow \nabla_{\mathbf{x}'_k} \left\| \mathbf{y} - \mathcal{R}(\mathbf{x}'_k) \right\|_2$
11:        $\mathbf{h}_k \leftarrow (\mathbf{g}_k - \mathbf{g}'_k)/\delta_k$
12:        $\alpha_k^{\mathrm{raw}} \leftarrow \left\| \mathbf{g}_k \right\|^2 / (\langle \mathbf{g}_k, \mathbf{h}_k \rangle + \varepsilon)$
13:        $\alpha_k \leftarrow \min\{\alpha_k^{\mathrm{raw}},\, 1/L\}$
14:        $\tilde{\mathbf{x}}_k \leftarrow \mathbf{x}_k - \alpha_k\, \mathbf{g}_k$
15:        **if** $\left\| \mathbf{y} - \mathcal{R}(\tilde{\mathbf{x}}_k) \right\|_2 > \left\| \mathbf{y} - \mathcal{R}(\mathbf{x}_k) \right\|_2 - \eta_{\mathrm{FCM}}\, \alpha_k \left\| \mathbf{g}_k \right\|^2$ **then**
16:          $\alpha_k \leftarrow \alpha_k / 2$     ▷ single Armijo back–off
17:          $\tilde{\mathbf{x}}_k \leftarrow \mathbf{x}_k - \alpha_k\, \mathbf{g}_k$
18:        **end if**
19:        $\mathbf{x}_{k+1} \leftarrow \tilde{\mathbf{x}}_k$     ▷ update iterate
20:     **end for**
21:     $\tilde{\boldsymbol{X}}_{0|t} \leftarrow \mathbf{x}_K$
    **(c) DDIM update**
22:     $\sigma_t \leftarrow \eta \sqrt{\frac{1 - \bar{\alpha}_{t-1}}{1 - \bar{\alpha}_t}} \sqrt{1 - \frac{\bar{\alpha}_t}{\bar{\alpha}_{t-1}}}$
23:     $\boldsymbol{\epsilon} \sim \mathcal{N}(\mathbf{0}, \mathbf{I})$
24:     $\boldsymbol{X}_{t-1} \leftarrow \sqrt{\bar{\alpha}_{t-1}}\, \tilde{\boldsymbol{X}}_{0|t} + \sqrt{1 - \bar{\alpha}_{t-1} - \sigma_t^2}\, \hat{\boldsymbol{\epsilon}}_t + \sigma_t\, \boldsymbol{\epsilon}$
25: **end for**
26: $\boldsymbol{X}_0 \leftarrow (\boldsymbol{X}_1 - \sqrt{1 - \bar{\alpha}_1}\, \boldsymbol{\epsilon}_{\theta^\star}(\boldsymbol{X}_1))/\sqrt{\bar{\alpha}_1}$
27: **return** $\boldsymbol{X}_0$

---

## A.2 Implementation Details

To model the reverse process $p_\theta$, we employed a Diffusion Transformer, originally introduced in Point-E's unconditional model [24], as the neural network parameterized by $\theta$, which predicts both $\mu_\theta$ and $\Sigma_\theta$. All images were set to a resolution of 224×224. For the ShapeNet dataset, 2,048 points were sampled and then upsampled to 8,192 points for comparison [27]. In the case of CO3D, 8,192 points were directly sampled. To sufficiently refine the point cloud, we perform four FCM updates per DDIM sampling step. We set the hyperparameters as follows: $\eta_{FCM} = 10^{-4}$, $L = 2/3$. For ShapeNet, we set $T = 256$ and $\delta_0 = 2 \times 10^{-2}$. For CO3D, we set $T = 512$ and $\delta_0 = 6 \times 10^{-3}$. We use point cloud rendering processes provided by PyTorch3D [26]. For ShapeNet, we set the radius of the point cloud rasterizer to 0.018 for airplane category and 0.027 for the other categories. For CO3D, we set the radius to 0.013. All experiments were performed using an NVIDIA RTX 6000 Ada Generation with a batch size of 16.

## A.3 Theoretical Analysis of FCM

In this appendix, we provide detailed proofs for the theoretical guarantees of our Forward Curvature-Matching (FCM) method. We begin by formally establishing the properties of FCM step sizes, followed by proofs of loss decrease and convergence guarantees. Finally, we analyze how FCM integrates with DDIM sampling while preserving its contraction properties.

### A.3.1 Bounds on FCM Step Sizes

We first establish that the FCM step size is guaranteed to lie within a well-behaved range, ensuring stable iterations. Our FCM approach relies on a directional curvature estimate:

$$\delta_k = \delta_0 \cdot \frac{\|\mathbf{x}_k\|}{\|\mathbf{g}_k\|}, \tag{14}$$

$$\mathbf{x}'_k = \mathbf{x}_k - \delta_k \cdot \mathbf{g}_k, \tag{15}$$

$$\mathbf{g}'_k = \nabla \mathcal{L}(\mathbf{x}'_k), \tag{16}$$

$$\mathbf{h}_k = \frac{\mathbf{g}_k - \mathbf{g}'_k}{\delta_k} \tag{17}$$

This $\mathbf{h}_k$ approximates the directional curvature along the gradient direction. Specifically, $\mathbf{h}_k$ is an approximation of the Hessian-vector product $\nabla^2 \mathcal{L}(\mathbf{x}_k)\mathbf{g}_k$.

**Lemma A.3.1** (Step Size Bounds). *Assume $\varepsilon \le L\|\mathbf{g}_k\|^2$ in the calculation of $\alpha_k^{raw}$. Then the FCM step size $\alpha_k$ (prior to any Armijo halving) satisfies:*

$$\frac{1}{2L} \le \alpha_k \le \frac{1}{L} \tag{18}$$

*Proof.* From the finite difference approximation with $\mathbf{h}_k = (\mathbf{g}_k - \mathbf{g}'_k)/\delta_k$, we analyze $\langle \mathbf{g}_k, \mathbf{h}_k \rangle$:

$$\langle \mathbf{g}_k, \mathbf{h}_k \rangle = \left\langle \mathbf{g}_k, \frac{\mathbf{g}_k - \mathbf{g}'_k}{\delta_k} \right\rangle \tag{19}$$

$$= \frac{1}{\delta_k}(\|\mathbf{g}_k\|^2 - \langle \mathbf{g}_k, \mathbf{g}'_k \rangle) \tag{20}$$

Under Assumption 3.1 ($L$-smoothness), we can establish that:

$$\langle \mathbf{g}_k, \mathbf{g}'_k \rangle \ge \|\mathbf{g}_k\|^2 - L\delta_k\|\mathbf{g}_k\|^2 \tag{21}$$

This implies:

$$\langle \mathbf{g}_k, \mathbf{h}_k \rangle \le \frac{1}{\delta_k}(\|\mathbf{g}_k\|^2 - (\|\mathbf{g}_k\|^2 - L\delta_k\|\mathbf{g}_k\|^2)) \tag{22}$$

$$= L\|\mathbf{g}_k\|^2 \tag{23}$$

Therefore:

$$\langle \mathbf{g}_k, \mathbf{h}_k \rangle + \varepsilon \le L\|\mathbf{g}_k\|^2 + \varepsilon \tag{24}$$

$$\le 2L\|\mathbf{g}_k\|^2 \tag{25}$$

where the last inequality holds given our assumption that $\varepsilon \le L\|\mathbf{g}_k\|^2$. This implies:

$$\alpha_k^{\text{raw}} = \frac{\|\mathbf{g}_k\|^2}{\langle \mathbf{g}_k, \mathbf{h}_k \rangle + \varepsilon} \ge \frac{1}{2L} \tag{26}$$

Since we cap $\alpha_k = \min\{\alpha_k^{\text{raw}}, 1/L\}$, we ensure $\alpha_k \le 1/L$ while maintaining the lower bound $\alpha_k \ge 1/(2L)$. $\qquad\square$

**Remark A.3.2.** *If $\|\mathbf{g}_k\| \approx 0$, the raw step size $\alpha_k^{raw}$ could become very large. However, in such cases, the Armijo condition will catch insufficient decrease and halve the step size once, still ensuring stable updates.*

### A.3.2 Firm Non-Expansiveness of the Gradient Step

Next, we establish that a gradient step with the FCM step size is firmly non-expansive, which is crucial for integrating with the diffusion process.

**Lemma A.3.3** (Firmly Non-Expansive Gradient Step). *Let $T_k(\mathbf{u}) = \mathbf{u} - \alpha_k \nabla \mathcal{L}(\mathbf{u})$ be the gradient step operator with fixed $\alpha_k > 0$. Under Assumptions 3.1–3.3, if $0 < \alpha_k < 2/L$, then $T_k$ is firmly non-expansive:*

$$\|T_k(\mathbf{u}) - T_k(\mathbf{v})\|^2 \leq \|\mathbf{u} - \mathbf{v}\|^2 - \alpha_k \left( \frac{2}{L} - \alpha_k \right) \|\nabla \mathcal{L}(\mathbf{u}) - \nabla \mathcal{L}(\mathbf{v})\|^2 \tag{27}$$

*Hence, $T_k$ is in particular non-expansive: $\|T_k(\mathbf{u}) - T_k(\mathbf{v})\| \leq \|\mathbf{u} - \mathbf{v}\|$.*

*Proof.* Let $\Delta = \mathbf{u} - \mathbf{v}$ and $\Delta_g = \nabla \mathcal{L}(\mathbf{u}) - \nabla \mathcal{L}(\mathbf{v})$. Then:

$$\|T_k(\mathbf{u}) - T_k(\mathbf{v})\|^2 = \|\Delta - \alpha_k \Delta_g\|^2 \tag{28}$$

$$= \|\Delta\|^2 - 2\alpha_k \langle \Delta, \Delta_g \rangle + \alpha_k^2 \|\Delta_g\|^2 \tag{29}$$

By the Baillon–Haddad theorem (which applies when $\mathcal{L}$ is convex and $L$-smooth), $\nabla \mathcal{L}$ is $1/L$-cocoercive, meaning:

$$\langle \Delta, \Delta_g \rangle \geq \frac{1}{L} \|\Delta_g\|^2 \tag{30}$$

Substituting this into our expression:

$$\|T_k(\mathbf{u}) - T_k(\mathbf{v})\|^2 \leq \|\Delta\|^2 - \frac{2\alpha_k}{L} \|\Delta_g\|^2 + \alpha_k^2 \|\Delta_g\|^2 \tag{31}$$

$$= \|\Delta\|^2 - \alpha_k \left( \frac{2}{L} - \alpha_k \right) \|\Delta_g\|^2 \tag{32}$$

Since $0 < \alpha_k \leq 1/L$ in our FCM algorithm (as established in Lemma A.3.1), the factor $\frac{2}{L} - \alpha_k > 0$. Thus, $T_k$ is firmly non-expansive, and consequently, $\|T_k(\mathbf{u}) - T_k(\mathbf{v})\| \leq \|\mathbf{u} - \mathbf{v}\|$. $\qquad\square$

### A.3.3 Guaranteed Loss Decrease

We now prove Theorem 3.4 from the main paper, which guarantees that FCM decreases the measurement loss at each iteration.

**Theorem A.3.4** (Guaranteed Loss Decrease). *Let $c = \min\{\frac{\eta_{FCM}}{2L}, \frac{1}{8L}\}$. The FCM algorithm ensures:*

$$\mathcal{L}(\mathbf{x}_{k+1}) \leq \mathcal{L}(\mathbf{x}_k) - c\|\nabla \mathcal{L}(\mathbf{x}_k)\|^2 \tag{33}$$

*Proof.* We consider two cases:

**Case 1 (Armijo condition satisfied):** When the initial step satisfies the Armijo condition, we have:

$$\mathcal{L}(\mathbf{x}_{k+1}) \leq \mathcal{L}(\mathbf{x}_k) - \eta_{\text{FCM}} \alpha_k \|\mathbf{g}_k\|^2 \tag{34}$$

$$\leq \mathcal{L}(\mathbf{x}_k) - \frac{\eta_{\text{FCM}}}{2L} \|\mathbf{g}_k\|^2 \tag{35}$$

where we used the lower bound $\alpha_k \geq \frac{1}{2L}$ from Lemma A.3.1.

**Case 2 (Armijo halving required):** If the initial step fails the Armijo condition and we halve $\alpha_k$, then $\alpha_k \geq \frac{1}{4L}$ remains. By the descent lemma for $L$-smooth functions:

$$\mathcal{L}(\mathbf{x}_{k+1}) \leq \mathcal{L}(\mathbf{x}_k) - \alpha_k \|\mathbf{g}_k\|^2 + \frac{L}{2} \alpha_k^2 \|\mathbf{g}_k\|^2 \tag{36}$$

$$= \mathcal{L}(\mathbf{x}_k) - \alpha_k \left( 1 - \frac{L\alpha_k}{2} \right) \|\mathbf{g}_k\|^2 \tag{37}$$

Since $\alpha_k \leq \frac{1}{2L}$ after halving, we have $1 - \frac{L\alpha_k}{2} \geq \frac{1}{2}$. Combined with $\alpha_k \geq \frac{1}{4L}$, this gives:

$$\mathcal{L}(\mathbf{x}_{k+1}) \leq \mathcal{L}(\mathbf{x}_k) - \frac{\alpha_k}{2}\|\mathbf{g}_k\|^2 \tag{38}$$

$$\leq \mathcal{L}(\mathbf{x}_k) - \frac{1}{8L}\|\mathbf{g}_k\|^2 \tag{39}$$

Taking the minimum of the guaranteed decrease in both cases, we get:

$$\mathcal{L}(\mathbf{x}_{k+1}) \leq \mathcal{L}(\mathbf{x}_k) - \min\left\{\frac{\eta_{\text{FCM}}}{2L}, \frac{1}{8L}\right\}\|\mathbf{g}_k\|^2 \tag{40}$$

$\square$

**Corollary A.3.5** (Gradient Norm Convergence). *Under FCM iterations, $\|\nabla\mathcal{L}(\mathbf{x}_k)\| \to 0$ as $k \to \infty$, and every cluster point is stationary.*

*Proof.* By Theorem A.3.4 and Assumption 3.2 (lower bound on $\mathcal{L}$), we have:

$$\sum_{k=0}^{\infty} c\|\nabla\mathcal{L}(\mathbf{x}_k)\|^2 \leq \mathcal{L}(\mathbf{x}_0) - \mathcal{L}_{\inf} < \infty \tag{41}$$

Since $c > 0$, we must have $\sum_{k=0}^{\infty}\|\nabla\mathcal{L}(\mathbf{x}_k)\|^2 < \infty$, which implies $\|\nabla\mathcal{L}(\mathbf{x}_k)\| \to 0$ as $k \to \infty$. This means that every cluster point of the sequence $\{\mathbf{x}_k\}$ is a stationary point of $\mathcal{L}$. $\square$

### A.3.4 FCM Integration with DDIM

Finally, we analyze how FCM integrates with the DDIM sampling process and prove Proposition 3.5 from the main paper.

**Proposition A.3.6** (Contraction Preservation). *Let $\Phi_t$ be a DDIM step that is $\rho$-contractive (with $\rho < 1$) in mean-square sense:*

$$\mathbb{E}[\|\Phi_t(\mathbf{u}) - \Phi_t(\mathbf{v})\|^2] \leq \rho\|\mathbf{u} - \mathbf{v}\|^2 \tag{42}$$

*Define $\Psi_t(\mathbf{u}) = T_k(\Phi_t(\mathbf{u}))$, where $T_k(\mathbf{u}) = \mathbf{u} - \alpha_k\nabla\mathcal{L}(\mathbf{u})$. Under Assumptions 3.1–3.3 and $\alpha_k \leq 1/L$, $\Psi_t$ is also $\rho$-contractive in expectation, thus preserving the diffusion contraction property.*

*Proof.* From Lemma A.3.3, we know that $T_k$ is non-expansive: $\|T_k(\mathbf{a}) - T_k(\mathbf{b})\|^2 \leq \|\mathbf{a} - \mathbf{b}\|^2$. Therefore, for any $\mathbf{u}, \mathbf{v}$:

$$\mathbb{E}[\|\Psi_t(\mathbf{u}) - \Psi_t(\mathbf{v})\|^2] = \mathbb{E}[\|T_k(\Phi_t(\mathbf{u})) - T_k(\Phi_t(\mathbf{v}))\|^2] \tag{43}$$

$$\leq \mathbb{E}[\|\Phi_t(\mathbf{u}) - \Phi_t(\mathbf{v})\|^2] \tag{44}$$

$$\leq \rho\|\mathbf{u} - \mathbf{v}\|^2 \tag{45}$$

Thus, $\Psi_t$ remains a $\rho$-contraction in mean-square sense. $\square$

### A.3.5 Robustness to Non-Convexity

While Assumption 3.3 (local convexity) is used in our theoretical analysis, FCM shows empirical robustness even when this assumption is violated.

**Remark A.3.7** (Behavior Under Non-Convexity). *If local convexity fails, the firm non-expansiveness of $T_k$ may break. However, Theorem A.3.4 and Corollary A.3.5 remain valid, guaranteeing that the FCM step decreases $\mathcal{L}$ and drives $\|\nabla\mathcal{L}(\mathbf{x}_k)\| \to 0$. This makes FCM robust in practice even for non-convex $\mathcal{L}$.*

### A.3.6 Practical Parameter Settings

Over-estimating $L$ in the algorithm only tightens the cap $\alpha_k \leq 1/L$ and preserves all theoretical guarantees. Under-estimating $L$ triggers the single Armijo halving, which prevents divergence while maintaining efficiency.

This combination of theoretical guarantees and practical robustness makes FCM an ideal choice for likelihood updates in diffusion-based 3D reconstruction, enabling high-quality results.

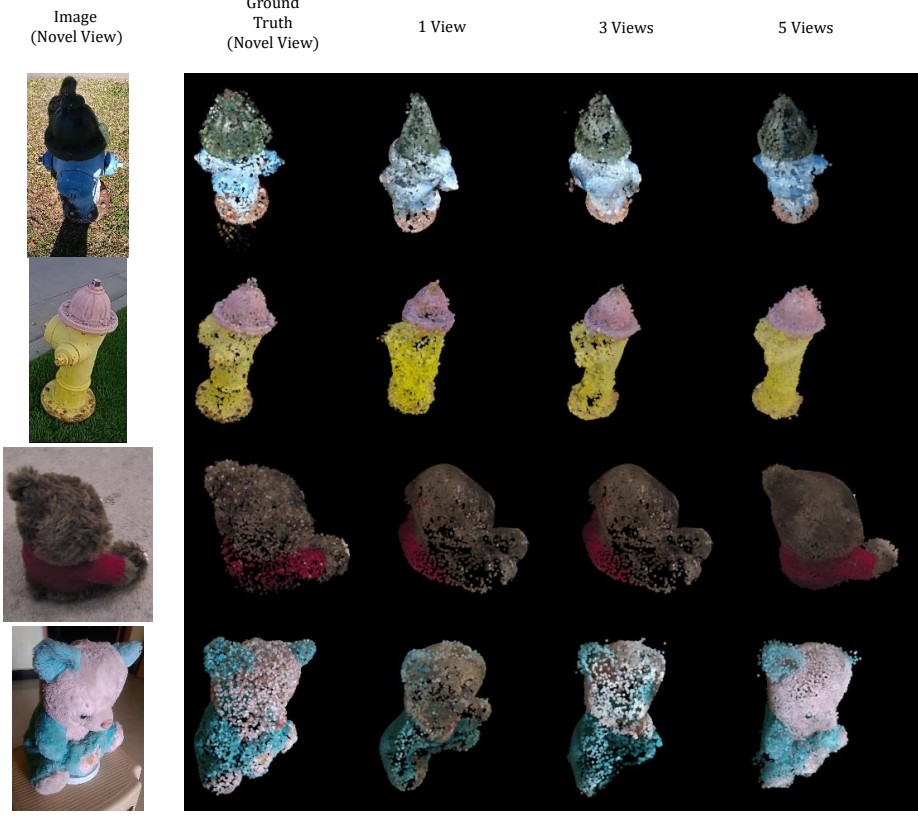

|  | Image
(Novel View) | Ground
Truth
(Novel View) | 1 View | 3 Views | 5 Views |

Figure 8: Qualitative results of multi view reconstruction on CO3D dataset.

| Category | EMD(×10) | | | CD(×10) | | | F-score | | | |
|---|---|---|---|---|---|---|---|---|---|---|
| | PC² [20] | BDM [34] | Ours | PC² [20] | BDM [34] | Ours | PC²† [20] | PC² [20] | BDM [34] | Ours |
| airplane | 0.551 | 0.552 | **0.476** | 0.434 | 0.409 | **0.378** | 0.473 | 0.457 | 0.524 | **0.543** |
| car | 0.524 | 0.535 | **0.517** | 0.487 | 0.507 | **0.460** | 0.359 | 0.331 | 0.330 | **0.386** |
| chair | **0.651** | 0.656 | 0.662 | **0.579** | 0.616 | 0.679 | 0.290 | 0.280 | 0.274 | **0.281** |
| table | 0.662 | **0.635** | 0.691 | 0.649 | **0.644** | 0.727 | 0.270 | 0.260 | 0.284 | **0.319** |
| *Average* | 0.597 | 0.594 | **0.587** | **0.542** | 0.544 | 0.561 | 0.348 | 0.332 | 0.353 | **0.382** |

Table 4: Extended comparison of single-view 3D reconstruction on ShapeNet. The table includes results from the original studies with about 1000 NFEs. Scores marked with † are reported from the original paper. Our method, however, achieves competitive performance with fewer function evaluations.

## A.4 Additional Experiments

**Comparison with other methods proposed in their original papers.** Tab. 4 shows a comparison of different models evaluated on the ShapeNet dataset, where each model is applied according to the method originally proposed in its respective study. PC² uses an NFEs of 1000, while BDM uses a total of 1080 NFEs—1000 for its reconstruction model and an additional 80 for its prior model.

| $L$ | EMD(×10) | CD(×10) | F-score | $\eta_{FCM}$ | EMD(×10) | CD(×10) | F-score | $\delta_0$ | EMD(×10) | CD(×10) | F-score |
|---|---|---|---|---|---|---|---|---|---|---|---|
| 100 | 0.731 | 0.770 | 0.307 | $10^{-6}$ | 0.585 | 0.563 | 0.373 | $5 \times 10^{-2}$ | 0.644 | 0.615 | 0.343 |
| 10 | 0.585 | 0.563 | 0.373 | $10^{-5}$ | 0.579 | 0.566 | 0.377 | $2 \times 10^{-2}$ | 0.587 | 0.561 | 0.382 |
| 1 | 0.588 | 0.564 | 0.376 | $10^{-4}$ | 0.587 | 0.561 | 0.382 | $10^{-2}$ | 0.594 | 0.574 | 0.369 |
| 2/3 | 0.587 | 0.561 | 0.382 | $10^{-3}$ | 0.586 | 0.565 | 0.370 | $10^{-3}$ | 0.665 | 0.660 | 0.330 |

Table 7: Hyperparameter study for FCM-guided sampling. Varying Lipschitz constant $L$, Armijo factor $\eta_{FCM}$ and the initial discrepancy radius $\delta_0$ for the scaled curvature probe.

| Average | EMD($\times 10$) | CD($\times 10$) | F-score |
|---------|---------|--------|---------|
| PC$^2$[20] | 2.662 | 3.893 | 0.244 |
| Ours | **1.206** | **1.527** | **0.281** |
| Ours(3-view) | 1.001 | 1.131 | 0.388 |
| Ours(5-view) | 0.941 | 1.020 | 0.423 |

Table 5: Quantitative results of CO3D dataset. The results represent the average values for two categories used in the experiments, with ground truth regularized to the range [-0.5, 0.5]. The F-score threshold is set to 0.2, and the CD corresponds to the results for the L1 metric.

| Method | EMD($\times 10$) | CD($\times 10$) | F-score | time($s/sample$) |
|--------|---------|--------|---------|---------|
| PC$^2$ [20] | 0.597 | **0.542** | 0.332 | 8.88 |
| BDM [34] | 0.594 | 0.544 | 0.353 | 10.56 |
| Ours | **0.587** | 0.561 | **0.382** | **6.03** |

Table 6: Time efficiency analysis of different methods. We report average scores of ShapeNet dataset and total sampling time (seconds per sample). The highest scores are marked in **bold**, and the second-highest scores are underlined. Our method achieves higher reconstruction accuracy with respect to F-score and EMD than prior approaches while maintaining comparable inference speed.

| Component | Ours | PC$^2$ | BDM |
|-----------|------|--------|-----|
| FCM update(1 iteration) | 5.339 ms | – | – |
| Local Conditioning | – | 3.486 ms | (same as PC$^2$) |
| NFEs | 256 | 1000 | 1080 |

| single Armijo check | strong Wolfe line search |
|---------------------|--------------------------|
| 0.889 ms | 6.589 ms |

Table 8: Runtime breakdown. Left: component costs and NFE counts for each method (ours: 256 NFEs; PC$^2$: 1000; BDM: 1080=1000+80). Right: cost of a single Armijo check (ours) versus a strong Wolfe line search. Our once-only Armijo strategy is substantially cheaper while preserving reliable descent, contributing to the overall speedup.

Despite using fewer NFEs, our method still achieves the highest scores in terms of both EMD and F-score.

**More experiments on CO3D dataset.** Tab. 5 presents a comparison with PC$^2$ on the CO3D dataset and additionally provides scores for the multi-view setting. These results demonstrate the improved performance of our method over existing approaches. Furthermore, Fig. 8 illustrates the qualitative results of multi-view reconstruction on CO3D. To demonstrate broader applicability, Fig. 10 presents qualitative comparisons on two additional CO3D categories—remote and vase—against PC$^2$.

**Hyperparameter Experiments.** As shown in Tab. 7, the sampler behaves robustly once each knob is kept within a reasonable range. In particular, as discussed in A.3.6, an overly conservative choice of $1/L$ (i.e., taking $L$ too large) activates the clamping in Eq. 11, so the effective update becomes overly damped and Armijo progress can stall, leading to slow or failed convergence. Nevertheless, the table shows substantial tolerance: the sampler remains reliable for $L \in [2/3, 10]$.

**Time analysis.** As shown in Tab. 6, our method achieves the best performance in both F-score and EMD score, highlighting its effectiveness even with a 32.1% reduction in runtime. As shown in Tab. 8, our once-only Armijo check is dramatically cheaper than a conventional strong Wolfe line search. While a single step in PC$^2$ or BDM can be faster than our FCM step, our sampler deliberately opts for far fewer steps: 256 NFEs versus 1000 (PC$^2$) and 1080 (BDM). This NFE gap dominates the end-to-end runtime—reducing the number of denoiser/forward evaluations—and leads to overall faster and more efficient reconstructions, without sacrificing accuracy.

**Qualitative comparison with DPS-based methods.** Fig. 9 shows a comparison between our method and different methods. Here, the step size for the DPS-based method is set to the optimal value of $\gamma = 0.05$ as identified in Fig. 7. While the DPS-based method captures the overall shape reasonably well, it fails to recover accurate color information. This limitation is attributed to the use of a fixed step size, which leads to suboptimal updates. In contrast, our method produces results that are more optimal with respect to the given measurements.

**More examples and failure cases.** Additional qualitative results are shown in Figs. 11- 18, and failure cases of our method are presented in Figs. 19- 21. Most failure cases occur when the object has a complex structure, which can be attributed to the diffusion prior being misled by unfamiliar or uncommon data distributions.

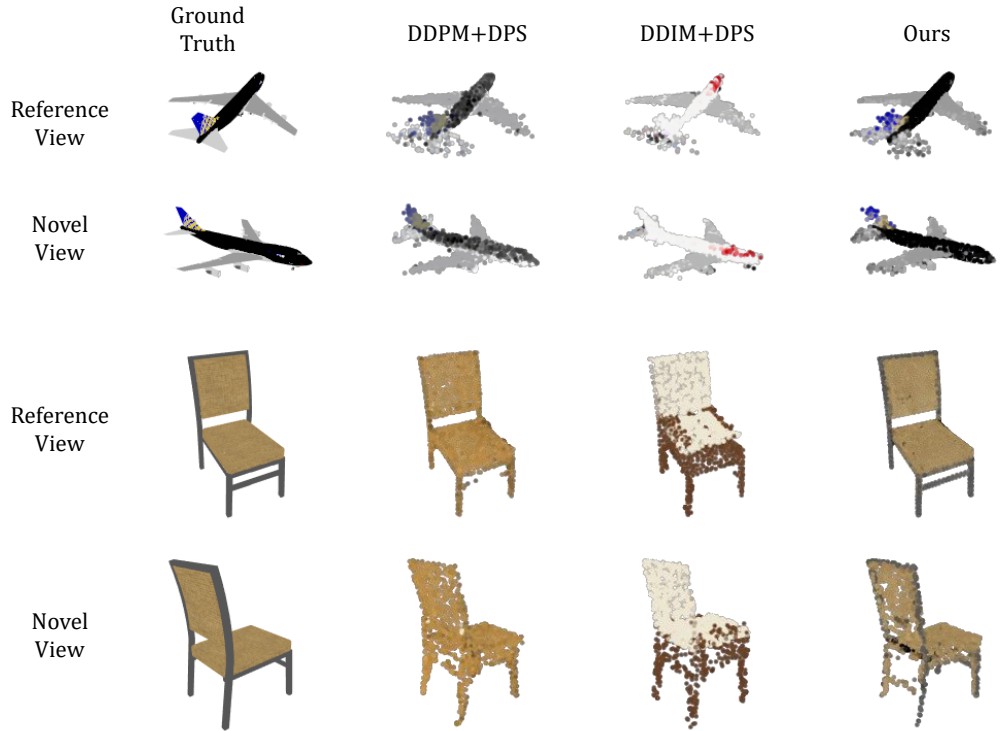

Figure 9: Qualitative comparison of reconstructions for different methods of sampling. Both of DPS based methods capture the overall shape but fail to preserve the correct colors due to the suboptimality.

## A.5 Limitations

- While the rendered image may resemble the reference image, the structure of the point cloud appears slightly thinner than the ground truth point cloud due to the radius of the rasterizer. This effect is particularly noticeable in thin structures, such as the legs of a chair. Additionally, due to the unavailability of a colored point cloud dataset for ShapeNet, we used a dataset generated by KeypointNet. However, since KeypointNet does not assign the actual mesh colors, this may lead to a degradation in the quality of the reconstruction.

- Direct control of point cloud positions for likelihood updates might limit CD metric performance compared to reconstructions using only the learned prior(or posterior). However, considering the objective of our task "image(s) to 3D reconstruction", metrics such as F-score and EMD, which measure the shape similarity, are more aligned with the task's purpose than measuring the distance between individual points.

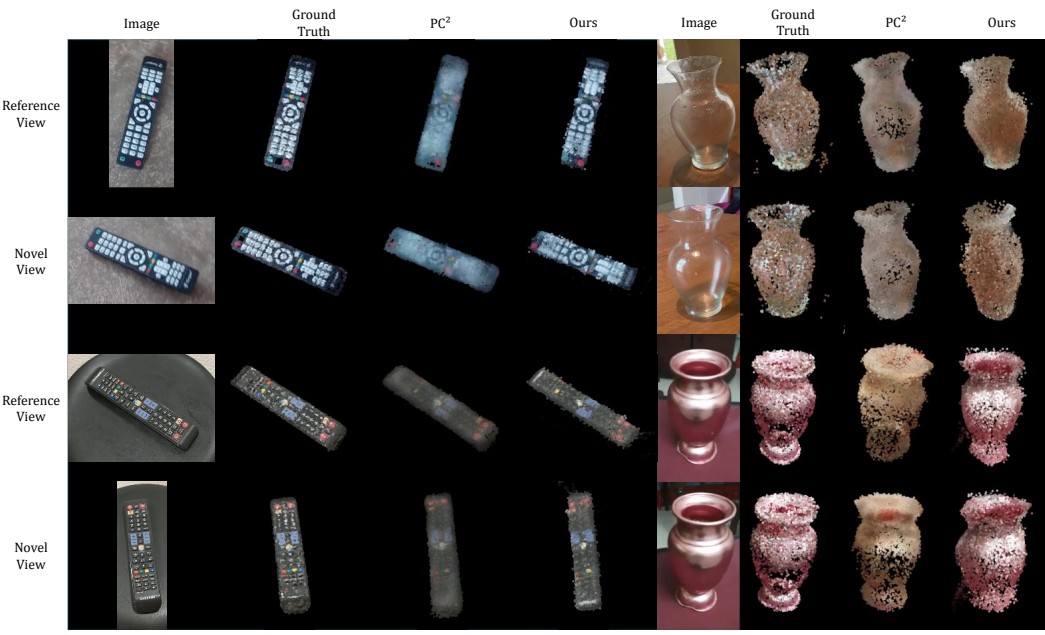

Figure 10: Qualitative comparison for single-view reconstruction on additional CO3D categories (remote, vase), against PC$^2$.

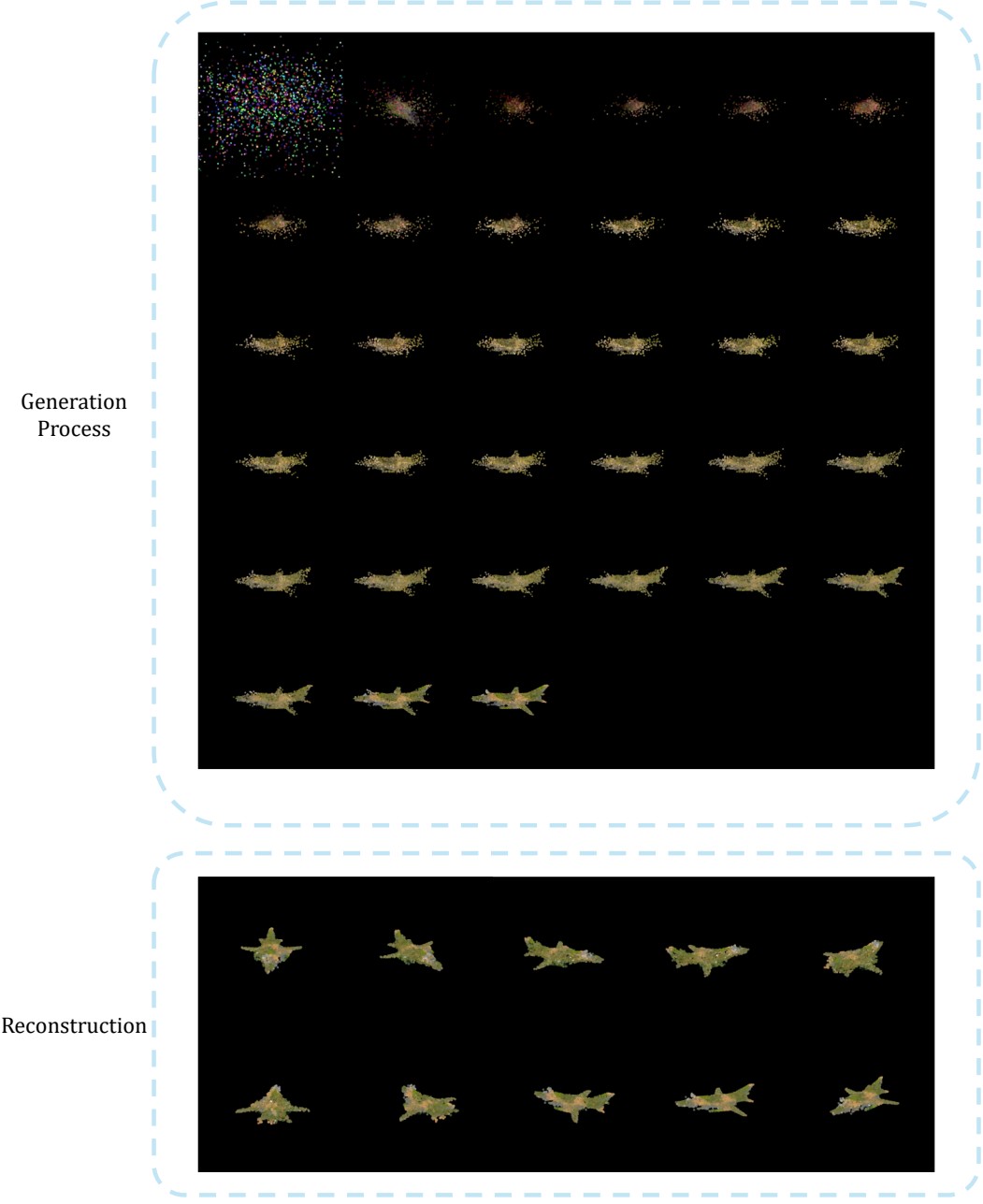

Figure 11: Generation trajectory and final reconstruction. Top: starting from pure noise at $T = 256$, the sampler progressively denoises toward a coherent airplane; we display every 8th diffusion level ($\Delta T = 8$) down to $T = 0$. Bottom: the resulting $T = 0$ sample rendered from multiple viewpoints.

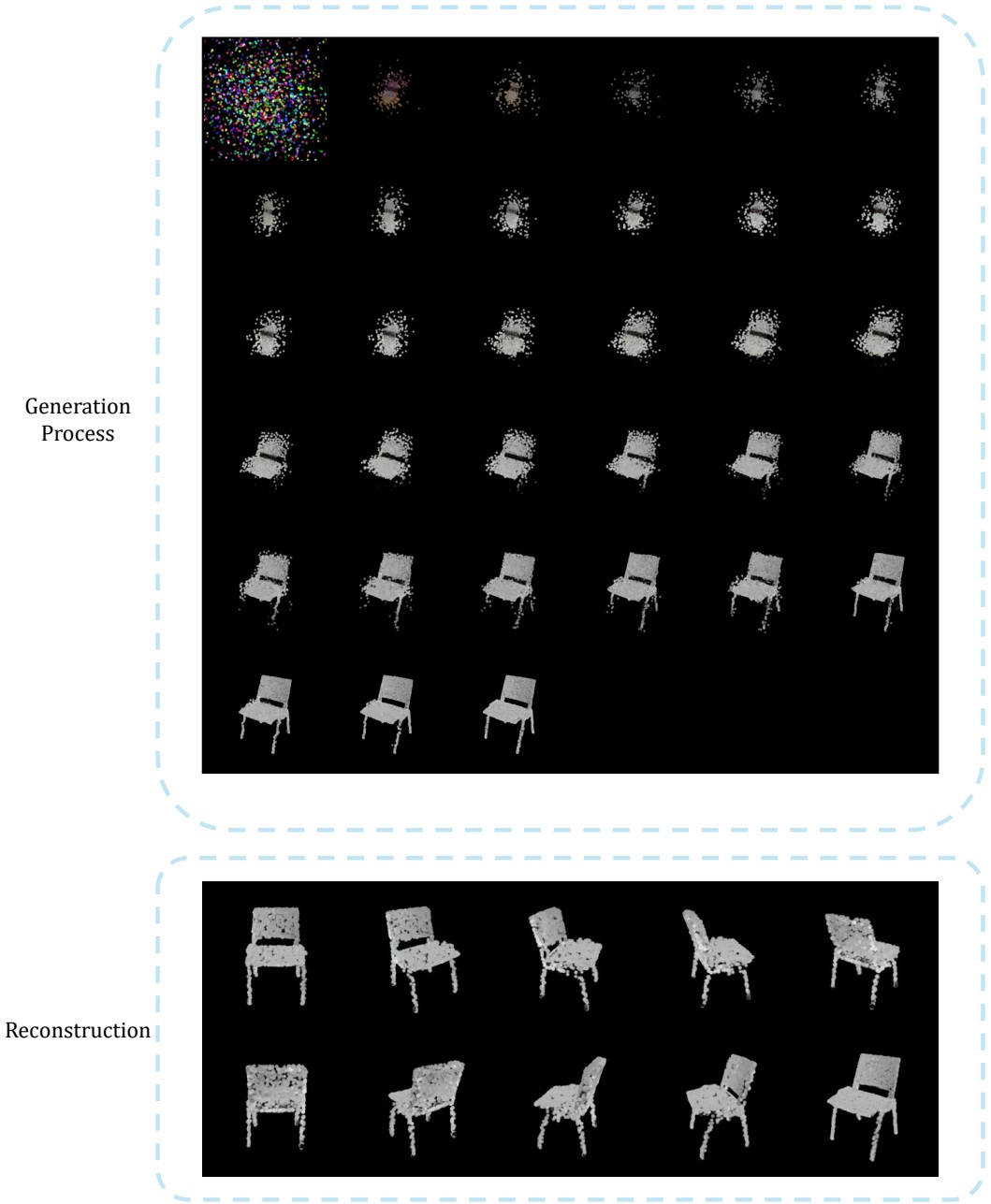

Figure 12: Generation trajectory and final reconstruction. Top: starting from pure noise at $T = 256$, the sampler progressively denoises toward a coherent chair; we display every 8th diffusion level ($\Delta T = 8$) down to $T = 0$. Bottom: the resulting $T = 0$ sample rendered from multiple viewpoints.

Reference
Image

Reference
View

Novel
Views

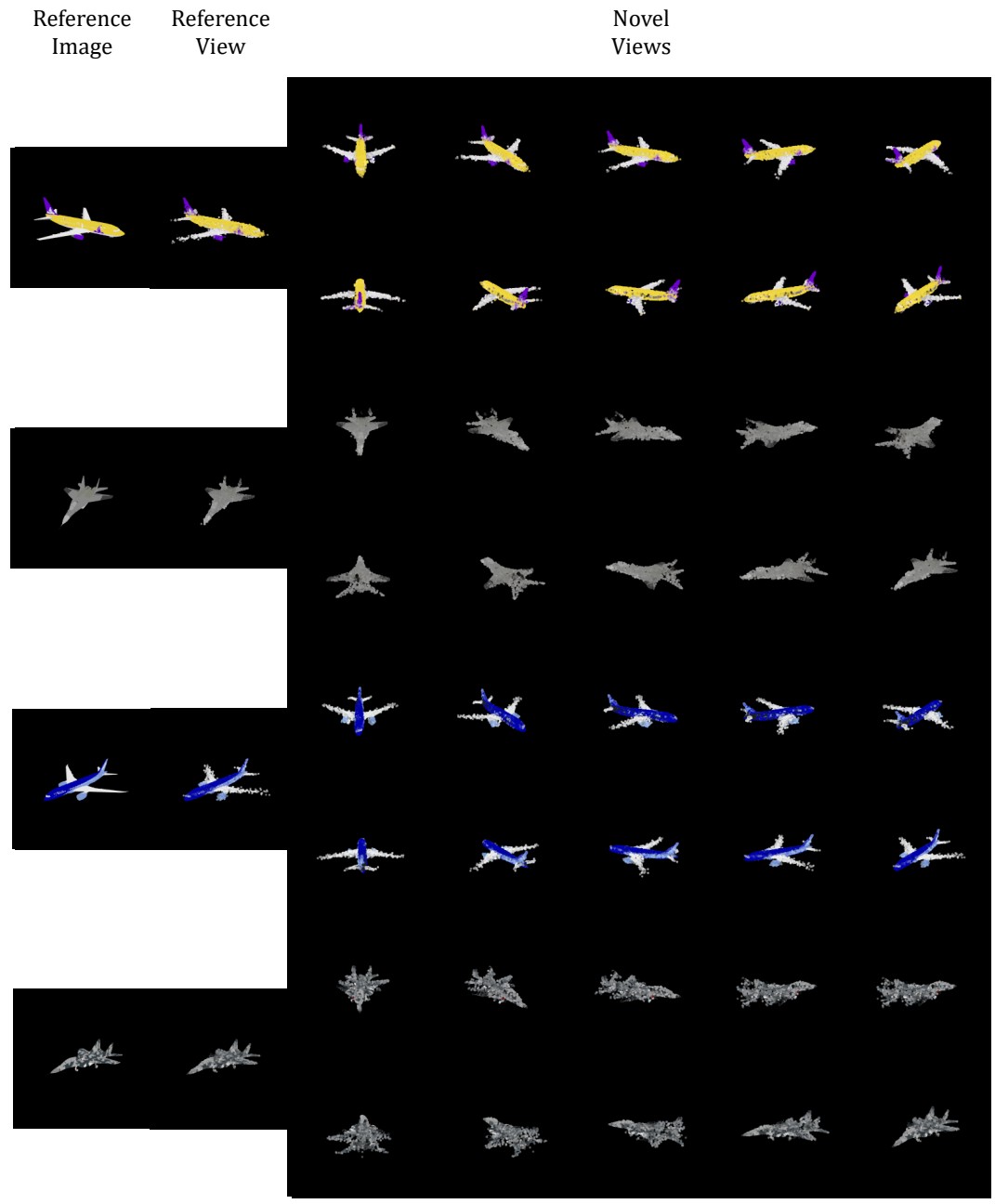

Figure 13: Additional qualitative results for single-view reconstruction on ShapeNet: Airplane

Reference
Image

Reference
View

Novel
Views

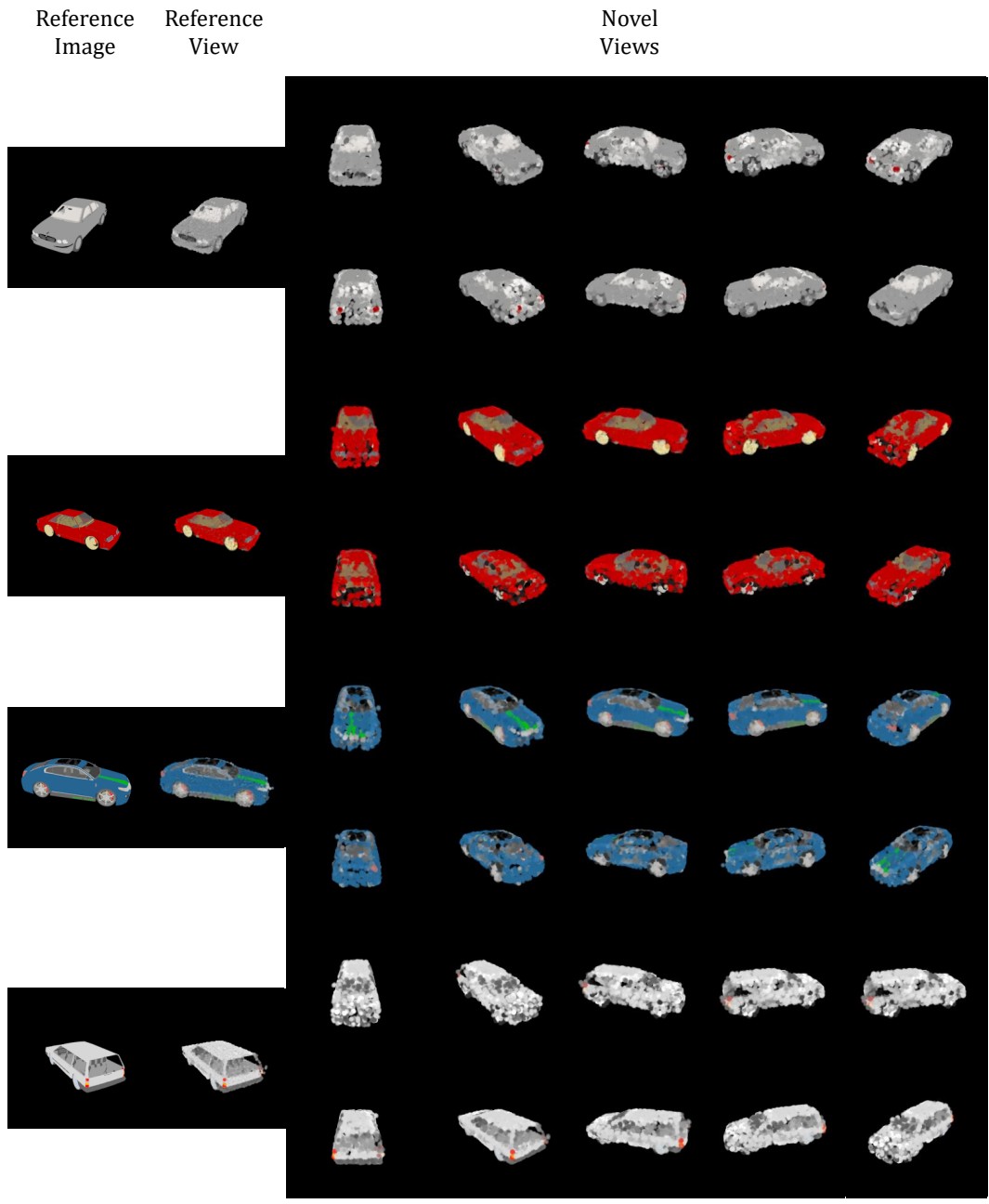

Figure 14: Additional qualitative results for single-view reconstruction on ShapeNet.: Car

Reference
Image

Reference
View

Novel
Views

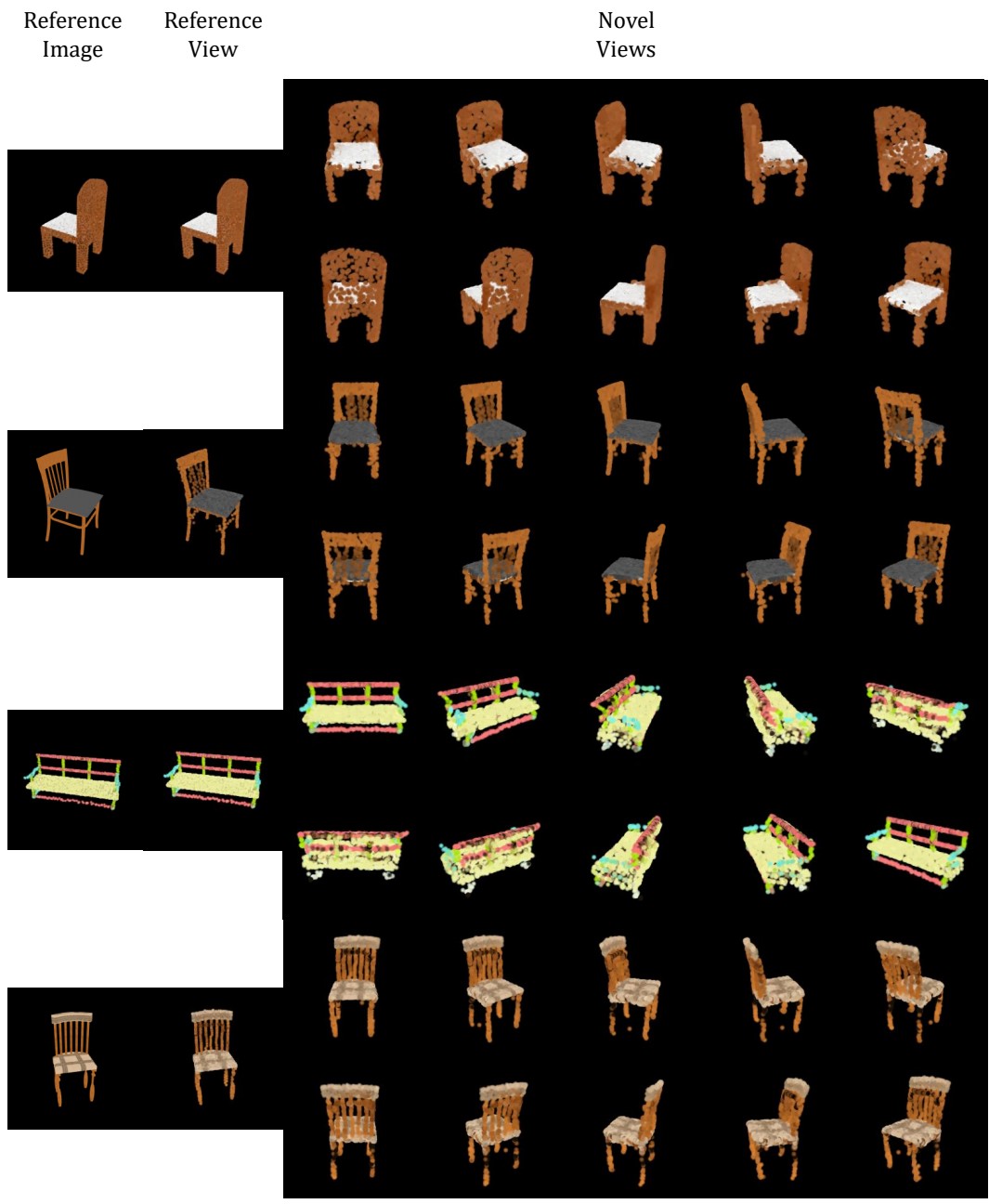

Figure 15: Additional qualitative results for single-view reconstruction on ShapeNet.: Chair

Reference
Image

Reference
View

Novel
Views

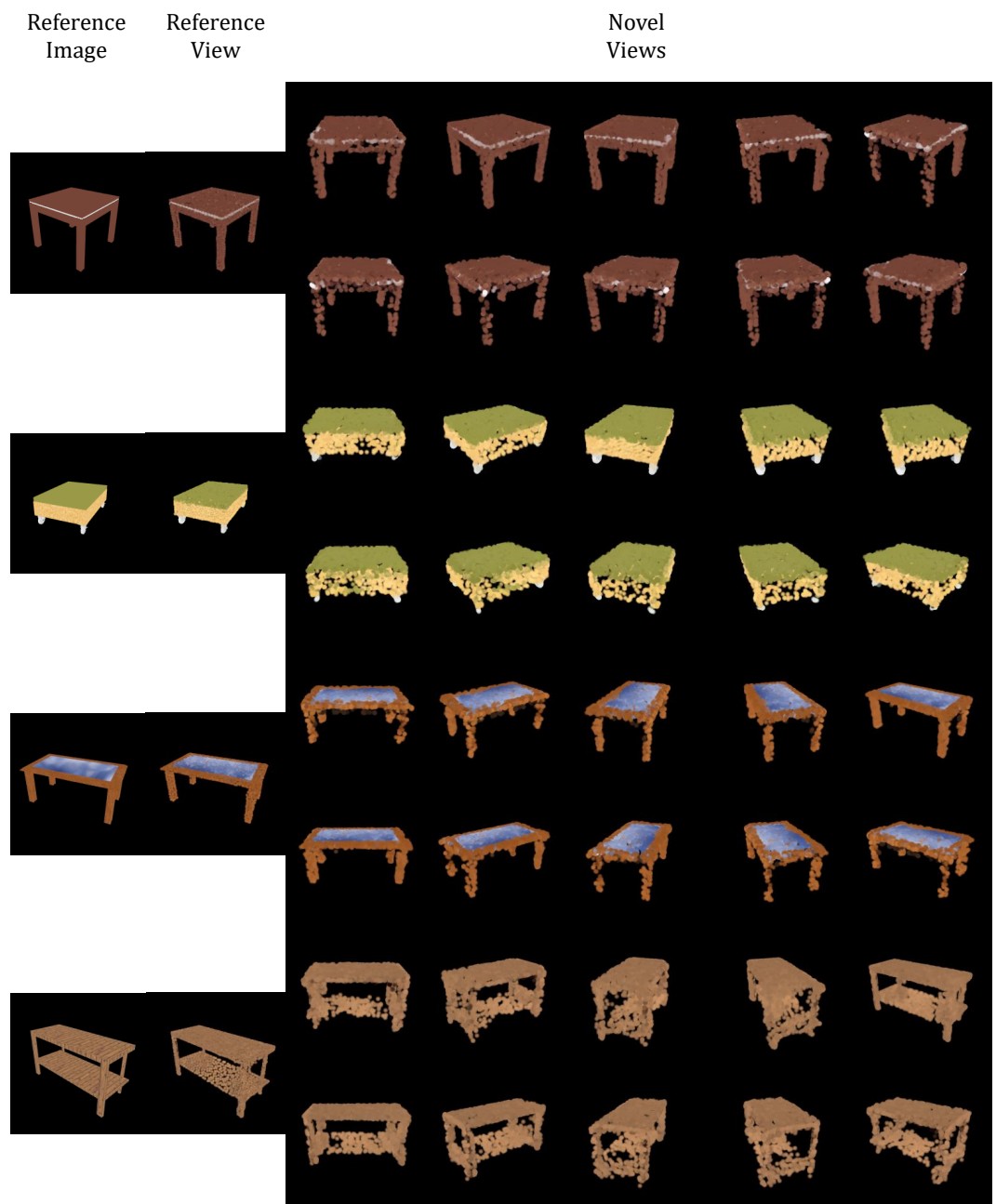

Figure 16: Additional qualitative results for single-view reconstruction on ShapeNet.: Table

Reference
Image

Reference
View

Novel
Views

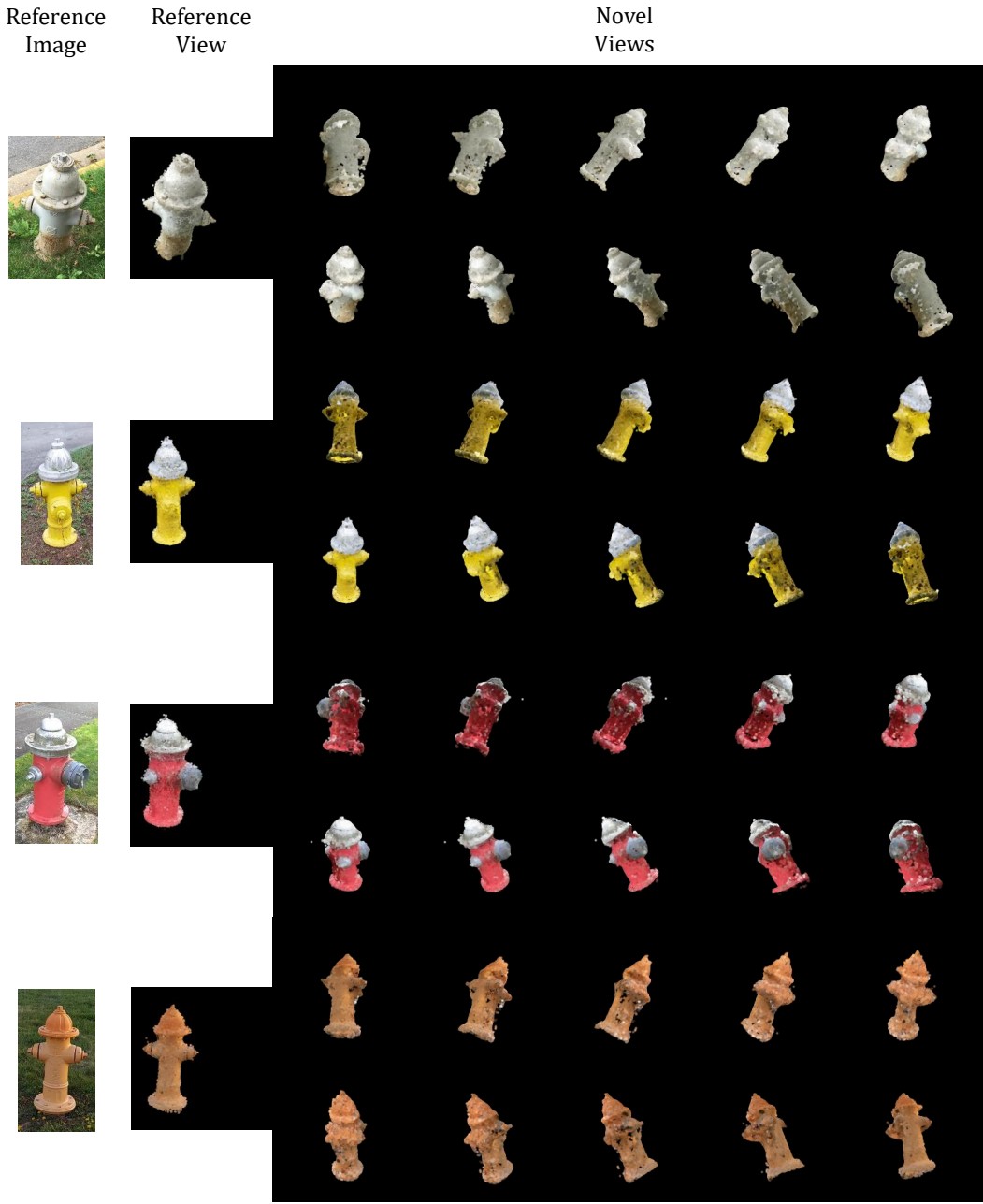

Figure 17: Additional qualitative results for single-view reconstruction on CO3D.: Hydrant

Reference
Image

Reference
View

Novel
Views

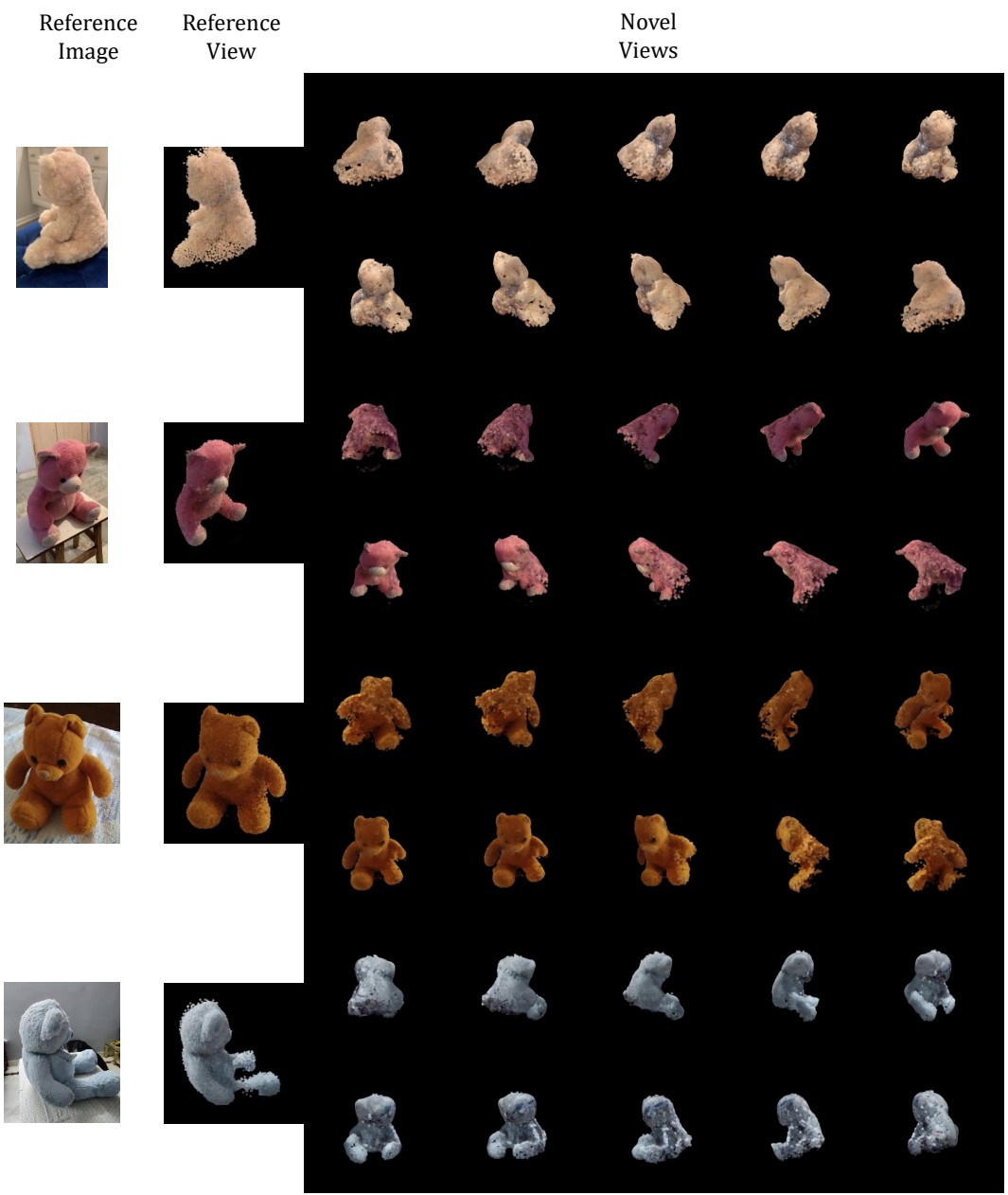

Figure 18: Additional qualitative results for single-view reconstruction on CO3D.: Teddybear

Reference
Image

Reference
View

Novel
Views

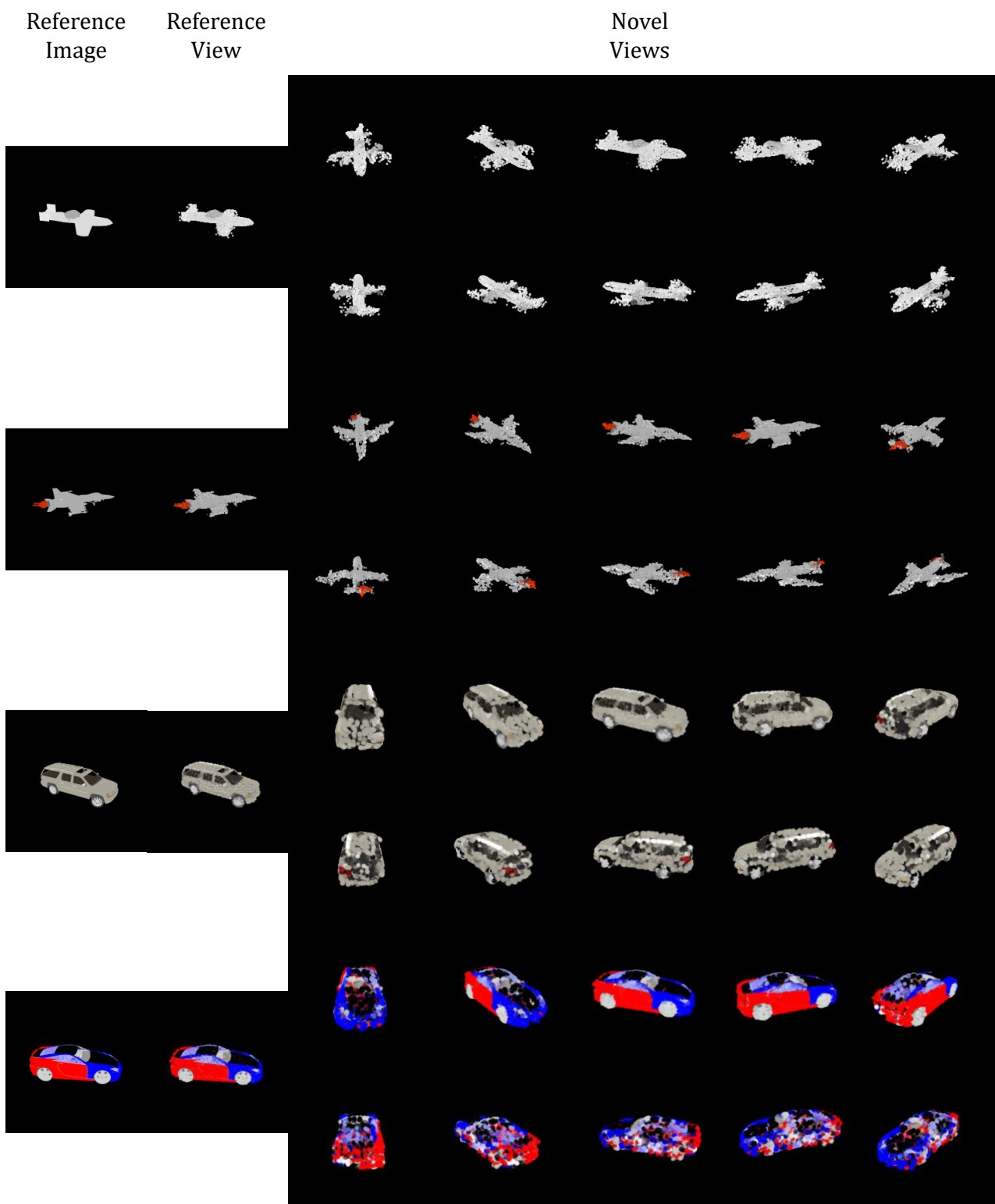

Figure 19: Analysis of failure cases in single-view reconstruction on ShapeNet.: Airplane&Car

Reference
Image

Reference
View

Novel
Views

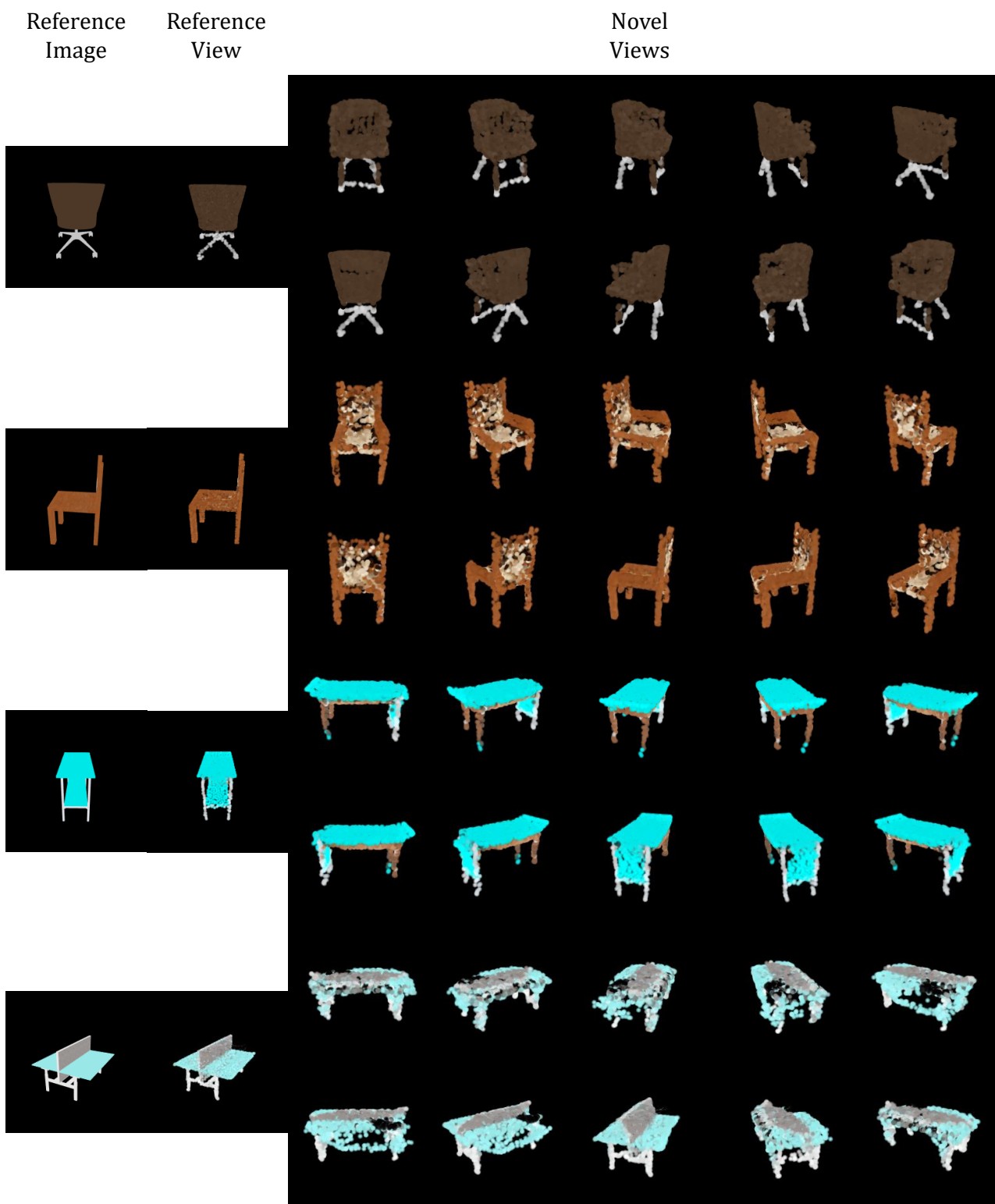

Figure 20: Analysis of failure cases in single-view reconstruction on ShapeNet.: Chair&Table

Reference
Image

Reference
View

Novel
Views

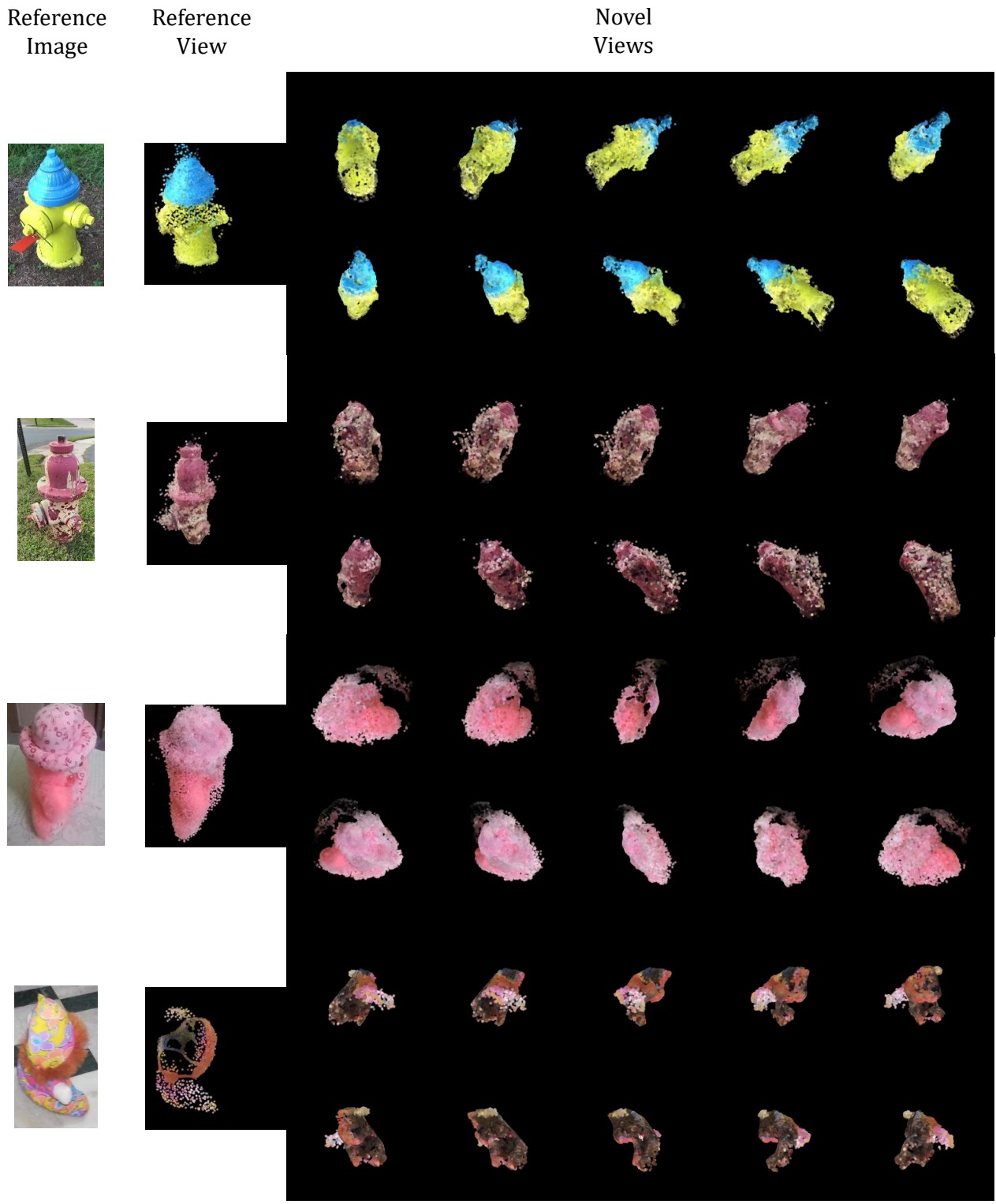

Figure 21: Analysis of failure cases in single-view reconstruction on CO3D.

