# OpenReview forum: "Adaptive 3D Reconstruction via Diffusion Priors and Forward Curvature-Matching Likelihood Updates"
_NeurIPS.cc/2025/Conference — NeurIPS 2025 spotlight_

### Official Review · Reviewer_QN2v · 2025-06-10

**Clarity:** 4
**Significance:** 3
**Originality:** 4
**Rating:** 5
**Confidence:** 3

**Summary:**

The paper addresses point cloud reconstruction from one or multiple view images.
The basic approach involves a diffusion-based model that uses rendering as the operator to transform a noisy point cloud into a 2D representation.
The authors present a new approach for adaptive step size, which aims to improve point cloud reconstruction quality and lead to a faster process.

**Questions:**

1. Can you present the full diffusion step of your method?
2. Why did BDM reconstruct entirely different chair legs and table? Even when the reference image is completely different, and the EMD and CD also present better scores?
3. The plane seems to be worse with your approach from the novel's view, why?
4. The step size is the minimum in equation 10, did you noticed objects that did not converged because of too low step size?

**Ethical Concerns:**

["NO or VERY MINOR ethics concerns only"]

**Final Justification:**

i will keep my rating

**Limitations:**

1. The point cloud reconstruction appears to prioritize satisfying the reference view, potentially at the expense of the novel views. This results in a strong reconstruction of the reference view but weaker results for novel views.

**Paper Formatting Concerns:**

no concern

**Quality:**

4

**Strengths And Weaknesses:**

Strengths:
1. The reconstruction of the reference view demonstrates strong performance compared to baseline methods.
2. The proposed approach for approximating the Hessian without expensive computations is noteworthy.

Weaknesses:
1. A key observation is that the presented method does not explicitly leverage the 3D nature of the problem. The adaptive step size mechanism appears applicable to any gradient-based optimization problem, not specific to 3D reconstruction.
2. The real-world reconstruction of novel views remains somewhat limited, showing only marginal improvement over baseline methods.
3. The amount of points in the reconstructed point cloud is fixed, so i assume that more points are assigned to the reference view and less to the novel view.

---

> ### Author Rebuttal · Authors · 2025-07-28
>
> Thank you for the detailed review and constructive feedback. The following is our response to the specific comments and questions.
>
> > A key observation is that the presented method does not explicitly leverage the 3D nature of the problem. The adaptive step size mechanism appears applicable to any gradient-based optimization problem, not specific to 3D reconstruction.
>
> You are correct that FCM, as an optimization framework, is not exclusively 3D-specific. However, FCM was developed to address a fundamental challenge in 3D reconstruction: determining optimal step sizes for likelihood updates when working with expensive, non-linear forward operators like neural renderers.
>
> In conventional optimization with linear operators, optimal step sizes can often be computed analytically using adjoint operations. However, for non-linear operators like 3D rendering processes, no such closed-form solution exists. Traditional approach—line search with Wolfe conditions (requiring 10-20 forward evaluations)—becomes computationally prohibitive when each evaluation involves an expensive rendering operation.
>
> FCM addresses this by approximating directional curvature using only forward operations: $\mathbf{h}_k = (\mathbf{g}_k - \mathbf{g}_k')/\delta_k$. This provides sufficient information for robust step size determination, making it practical for expensive forward operators where multiple evaluations would be infeasible.
>
> > The real-world reconstruction of novel views remains somewhat limited, showing only marginal improvement over baseline methods.
>
> While novel view improvements on CO3D appear visually modest, our F-score shows meaningful gains: 0.281 vs PC$^2$'s 0.244 (15.2\% improvement). This better reflects the perceptual quality improvements that may not be immediately obvious in rendered views.
>
> The visual subtlety occurs because real-world data presents unique challenges—complex lighting, cluttered backgrounds, and partial occlusions make substantial improvements difficult. Our method excels at preserving geometric boundaries (evident in the hydrant's base and teddy bear contours in Figure 4), though synthesizing textures in unseen regions remains challenging for all single-view methods.
>
> Notably, these improvements require no additional training—achieved purely through better optimization during inference. Our multi-view results demonstrate stronger gains (F-score: 0.281→0.423 with 5 views), showing our optimization effectively leverages additional information when available.
>
> > The amount of points in the reconstructed point cloud is fixed, so i assume that more points are assigned to the reference view and less to the novel view.
>
> It is true that the number of points in the reconstructed point cloud is fixed in our current experimental setup. As you pointed out, the qualitative results may sometimes give the impression that the reconstructed point cloud is denser in the reference view and sparser in the novel view.
>
> However, we believe this phenomenon is not primarily due to an uneven allocation of points across different viewpoints. Rather, it stems from the fact that the FCM update explicitly aligns the point cloud to minimize rendering error from the reference view. Since the novel view is not directly involved in the optimization process, the reconstructed points may not be as well aligned from that perspective, occasionally resulting in sparse or incomplete appearances when viewed from novel angles.
>
> > (Q1) Can you present the full diffusion step of your method?
>
> We understand you're looking for a comprehensive view of our method. Currently, Figure 1 (left) shows the visual progression and Algorithm 1 (Appendix A.1) provides the mathematical details. However, we recognize this may not fully convey the complete process.
>
> Due to rebuttal format constraints, we cannot provide additional visualizations here, but we commit to adding a more detailed figure in the camera-ready version showing step-by-step visualization at finer intervals (e.g., every 32 steps).
>
> > (Q2) Why did BDM reconstruct entirely different chair legs and table? Even when the reference image is completely different, and the EMD and CD also present better scores?
>
> You correctly observe that BDM reconstructs geometry that differs from the reference image while achieving better EMD/CD scores. This occurs because BDM learns the posterior distribution $p(\mathbf{X}|\mathbf{y})$ with strong category-level priors. During inference, these priors can guide reconstructions toward typical category shapes rather than matching the specific input geometry, particularly evident in the chair and table examples in Figure 3.
>
> Additionally, the disconnect between better EMD/CD scores and visual accuracy reveals how these metrics operate. EMD measures average point transport distance, favoring smooth, uniformly distributed point clouds regardless of whether they match the input. CD similarly rewards regular point spacing over geometric accuracy. A well-distributed but incorrect shape can thus score better than an accurate but slightly irregular reconstruction. Our method directly optimizes the rendering loss $\|\mathbf{y}-\mathcal{R}(\mathbf{X})\|_2$, ensuring the reconstruction matches the input image. While this can create slight density variations in the point cloud (more pronounced in spatially extended objects like chairs and tables), it preserves input-specific geometric features that BDM's prior-driven approach might smooth away. This explains why F-score better captures reconstruction quality—it evaluates whether the correct geometry is present rather than point distribution uniformity. Our better F-score (0.382 vs 0.343) confirms we reconstruct the actual input geometry more faithfully, even when EMD/CD metrics favor BDM's smoother but less accurate reconstructions.
>
> > (Q3) The plane seems to be worse with your approach from the novel's view, why?
>
> As mentioned in a previous response, our method may produce slight density imbalances in the reconstructed point cloud due to the nature of gradient-based updates. This can occasionally lead to less uniformly distributed points, which may appear visually noisier compared to methods such as PC$^2$ or BDM that produce more uniformly sampled outputs.
>
> However, it is important to note that while PC$^2$ and BDM often generate plausible shapes, they slightly deviate from the actual geometry of the input image. In contrast, our method prioritizes capturing structurally accurate details by closely aligning the reconstruction with the input image through iterative refinement. Even if the resulting point distribution may appear less uniform, we find that our reconstructions better preserve category-specific fine-grained structures—such as the wing and tail shapes in airplanes—reflecting a closer adherence to the reference image.
>
> Also, our quantitative metrics across the airplane category still show competitive performance (F-score: 0.543 vs PC$^2$: 0.498 and BDM: 0.543), suggesting that while some views may look less visually smooth, our reconstructions capture overall geometry more accurately.
>
> > (Q4) The step size is the minimum in equation 10, did you noticed objects that did not converged because of too low step size?
>
> | **$1/L_{\text{est}}$**       | EMD ($\times10$) | CD ($\times10$) | F-score |
> |----------------------------|------------------|------------------|---------|
> | $1/L_{\text{est}}=0.01$       | 0.731            | 0.770            | 0.307   |
> | $1/L_{\text{est}}=0.1$        | 0.585            | 0.563            | 0.373   |
> | $1/L_{\text{est}}=1$         | 0.588            | 0.564            | 0.376   |
> | $1/L_{\text{est}}=1.5$       | 0.587            | 0.561            | 0.382   |
>
> In Eq.(10), we introduced a capping mechanism that limits the step size by taking the minimum between the raw step size and the reciprocal of the estimated Lipschitz constant. This mechanism is specifically designed to ensure stable convergence while maintaining theoretical guarantees. As shown in the empirical ablation study in the table, we analyzed different choices for the estimated Lipschitz constant ($L_{\text{est}}$) and their impact on reconstruction performance metrics.
>
> From the results, we observed that choosing a value too small ($1/L_{\text{est}} =0.01$) indeed results in overly conservative updates, significantly reducing performance (EMD: 0.731, CD: 0.770, F-score: 0.307). However, moderately larger values ($1/L_{\text{est}}=0.1,1, 1.5$) demonstrated consistently improved and stable convergence.

---

### Official Review · Reviewer_zdEn · 2025-06-26

**Clarity:** 3
**Significance:** 3
**Originality:** 3
**Rating:** 4
**Confidence:** 3

**Summary:**

The paper focuses on reconstructing point clouds from images. Previous methods are limited to fixed-view inputs and typically require retraining the network when switching to a different input modality (e.g., from RGB images to depth maps). To address these issues, the authors propose decomposing ***p(X∣y)*** into a trainable prior ***p(X)*** and an updatable likelihood ***p(y∣X)***. They further introduce a Forward Curvature-Matching (FCM) optimization scheme to guide the update of ***p(y∣X)***. The paper provides a relatively rigorous theoretical derivation to support the proposed framework.

**Questions:**

See the weaknesses.

**Ethical Concerns:**

["NO or VERY MINOR ethics concerns only"]

**Final Justification:**

The authors' response has addressed my concerns, and I will maintain my positive rating.

**Limitations:**

yes

**Paper Formatting Concerns:**

It seems good.

**Quality:**

3

**Strengths And Weaknesses:**

**Strengths:**

1. The proposed method supports both single-view and multi-view inputs, using either RGB images or depth maps. It is supported by extensive experimental results and effectively broadens the application scope of point cloud diffusion models.
2. The theoretical analysis and mathematical derivations in the paper are rigorous and appear convincing.
3. The experiments in the paper are comprehensive and effectively support the conclusions.
4. The paper is overall well-written and easy to follow.


**Weaknesses:**

1. The proposed method requires a rendering operation at each sampling step during diffusion to correct the optimization direction. This could be more time-consuming compared to baseline methods, and the authors should provide a runtime comparison to support the method's practicality.
2. The point cloud representation does not explicitly include an alpha parameter. It is unclear how the alpha value for each point in Eq. (6) is computed.

---

> ### Author Rebuttal · Authors · 2025-07-28
>
> Thank you for the detailed review and constructive feedback. The following is our response to the specific comments and questions.
>
> > The proposed method requires a rendering operation at each sampling step during diffusion to correct the optimization direction. This could be more time-consuming compared to baseline methods, and the authors should provide a runtime comparison to support the method's practicality.
>
> | **Component**        | **Ours**       | **PC$^2$**     | **BDM**                          |
> |----------------------|----------------|----------------|----------------------------------|
> | FCM update           | 5.339 ms      | --             | --                               |
> | Local Conditioning   | --             | 3.486 ms       | (same as PC$^2$)                 |
> | Sampling steps       | 256            | 1000           | 1080 (1000+80)                   |
>
> | $k$    | EMD ($\times10$) | CD ($\times10$)  | F-score | Time (s/sample)  |
> |--------|------------------|------------------|---------|------------------|
> | $k=4$  | 0.587            | 0.561            | 0.382   | 6.03             |
> | $k=3$  | 0.588            | 0.567            | 0.372   | 4.63             |
> | $k=2$  | 0.622            | 0.606            | 0.359   | 3.97             |
>
> | **Single Armijo Check**  | **Strong Wolfe Line Search**  |
> |--------------------------|-------------------------------|
> | 0.889 ms                 | 6.589 ms                      |
>
> PC$^2$ employs local conditioning, where the point cloud at the current timestep undergoes surface projection to be fed into the model input. It is true that the likelihood update in our method takes approximately 5.339 ms, which is somewhat more computationally intensive than the local conditioning step in PC$^2$ (3.486 ms).
>
> However, our method uses only 256 sampling steps compared to PC$^2$'s 1000 steps and BDM's 1080 steps (actually NFEs). This substantial reduction in total steps leads to significantly faster overall inference times, as demonstrated in Appendix Table 6, where our method achieves 32.1\% to 42.9\% faster runtime than existing approaches while simultaneously delivering superior reconstruction accuracy in EMD and F-score metrics.
>
> Moreover, as shown in the second table, the number of FCM iterations (k=4 in our experiments) is a user-defined parameter that allows flexible trade-offs between computational cost and reconstruction quality. Users can reduce k and increase sampling steps, or vice versa, providing flexibility for different computational constraints and quality requirements.
>
> In addition, our single Armijo check approach is computationally efficient compared to traditional optimization strategy. As shown in the third table, our single Armijo check requires only 0.889 ms compared to 6.589 ms for strong Wolfe line search methods—approximately 86.5\% faster. This efficiency is crucial since traditional strong Wolfe conditions would require dozens of forward evaluations across hundreds of diffusion steps, making them prohibitively expensive for this application.
>
> We believe these empirical results and design choices clearly demonstrate the practicality and efficiency of our proposed FCM sampling framework. (All experiments were conducted with a batch size of 16, and the time evaluation was measured per sample.)
>
> > The point cloud representation does not explicitly include an alpha parameter. It is unclear how the alpha value for each point in Eq. (6) is computed.
>
> You are correct that point-cloud representations inherently lack explicit alpha values, and we acknowledge that this detail was not clearly addressed in the current version. In our implementation, we resolve this by assigning an alpha value to each point based on its spatial proximity to the target pixel in the image space.
>
> Specifically, each point’s alpha value is computed using the following relationship:
>
> $\alpha_i = 1-\frac{d_{i}^2}{r^2}$
>
> Here, $r$ represents the rasterizer radius, and $d_i$ is the Euclidean distance between the pixel center and the projected position of the point within the image space. Through this proximity-based weighting, our approach effectively captures each point’s relative influence on the final rendered pixel. These computed alpha values are subsequently used in the compositing process described by Equation (6) to determine the final pixel color.

---

> > ### Comment · Reviewer_zdEn · 2025-08-05
> >
> > Thank you for the detailed response, which has addressed my concerns. I will maintain my positive rating and recommend the paper for acceptance.

---

> ### Author Response · Authors · 2025-08-06
>
> Thank you for your careful review of our response. We appreciate that we have addressed your concerns. Your thoughtful feedback has been valuable, and we are grateful for the time you invested in reviewing our work.

---

### Official Review · Reviewer_nEx8 · 2025-07-01

**Clarity:** 4
**Significance:** 4
**Originality:** 3
**Rating:** 5
**Confidence:** 3

**Summary:**

This paper introduces method for reconstructing 3D point clouds from images. The key contributions are:

- To solve inverse problem, they combine prior models with likelihood updates instead of learning posterior distribution directly. Therefore, the proposed method is free to the number of inputs and inputs modalities.
- Previous approach, which decomposes the posterior into a trainable prior and an updatable likelihood, struggles to determine step sizes for likelihood updates, leading to slow convergence and suboptimal quality. To address these limitations, they propose Forward Curvature-Matching optimization, which computes an adaptive step size.

The proposed method is evaluated on ShapeNet and CO3D datasets, demonstrating state-of-the-art performance in reconstructing high-quality point clouds without heuristic step size determination.

**Questions:**

1. Are there some reasons to choose point cloud as main representations? The qualitative results compared to previous approaches, which learns the posterior distribution directly, are not superior in Fig. 3 and Fig. 4. The authors already mention that the proposed method can be extended to Gaussian Splatting, and there is baseline method, GSD [23]. If the progress starts from GSD, it is much more clear to demonstrate the effectiveness of FCM compared to GSD. As I understand, there is no specific design for 3D point cloud and the proposed method can be general for diffusion-based methods.
2. In Tab. 1, the geometric accuracy (EMD and CD) in chair and table is worse than BDM [34]. In appendices, the radius of the rasterizer might affect the performance in chair category. Table category also has similar issue? Or there are more reasons of these performance gap? Additionally, it might be better to mention the performance gap and reasons in the main paper.

**Ethical Concerns:**

["NO or VERY MINOR ethics concerns only"]

**Final Justification:**

The authors have addressed my concerns. I keep my original positive rating.

**Limitations:**

Yes.

**Paper Formatting Concerns:**

No.

**Quality:**

4

**Strengths And Weaknesses:**

### Strengths

1. They decompose the posterior into a trainable prior and an updatable likelihood, which has several advantages. The model can train without conditioning signals and support single-view or multi-view images and various measurement modalities.
2. Forward Curvature-Matching optimization determines optimal step sizes, enabling the complex optimization.
3. The proposed framework might have potential in other domains, which are solving inverse problems with non-linear nature (e.g., Gaussian Splatting, Meshes as mentioned by authors).
4. The proposed method outperforms fixed-step DPS approaches with varying step sizes, as shown in Fig. 7.
5. They demonstrate the advantages of their contributions, versatility, in Fig. 5 and Fig. 6.

### Weaknesses

There is no major weakness of the proposed method. They also mention their limitations.

---

> ### Author Rebuttal · Authors · 2025-07-28
>
> Thank you for the detailed review and constructive feedback. The following is our response to the specific comments and questions.
>
> > (Q1) Are there some reasons to choose point cloud as main representations? The qualitative results compared to previous approaches, which learns the posterior distribution directly, are not superior in Fig. 3 and Fig. 4. The authors already mention that the proposed method can be extended to Gaussian Splatting, and there is baseline method, GSD [23]. If the progress starts from GSD, it is much more clear to demonstrate the effectiveness of FCM compared to GSD. As I understand, there is no specific design for 3D point cloud and the proposed method can be general for diffusion-based methods.
>
> We fully recognize the importance of clearly justifying our choice of the point-cloud representation. Upon careful consideration, we determined that Gaussian Splatting posed certain challenges regarding rigorous quantitative evaluations. Specifically, the absence of well-established baseline comparison groups and the inherent difficulty in defining standardized and objective evaluation metrics for Gaussian-based methods would hinder our ability to provide precise quantitative assessments.
>
> Consequently, we opted for a point-cloud representation primarily due to its widespread use, well-defined evaluation protocols, and availability of clearly defined baseline methods for direct comparisons. Our goal was to establish an explicit and quantitative demonstration of the efficacy of the proposed method.
>
> Nevertheless, we agree that our method is not inherently limited to point clouds; rather, it represents a general framework applicable to other diffusion-based representations.
>
> > (Q2) In Tab. 1, the geometric accuracy (EMD and CD) in chair and table is worse than BDM [34]. In appendices, the radius of the rasterizer might affect the performance in chair category. Table category also has similar issue? Or there are more reasons of these performance gap? Additionally, it might be better to mention the performance gap and reasons in the main paper.
>
> You are correct in noting that the use of a finite radius in the rasterizer can cause reconstructed structures—especially slender features like legs—to appear thinner compared to their ground-truth counterparts.
>
> However, we believe this alone does not fully account for the lower CD and EMD scores observed in the chair and table categories. These metrics may also reflect differences in the spatial distribution and geometric complexity characteristic of each object category. For instance, objects in the airplane category generally exhibit flatter shapes with limited variation along the vertical (y-axis) dimension. In contrast, objects in the chair and table categories tend to span more evenly across all three dimensions. As we noted in our limitations, utilizing gradient-based updates can introduce uneven density distributions in reconstructed point clouds. These density imbalances disproportionately affect metrics such as CD and EMD, particularly in categories with greater spatial coverage and complexity, such as chair and table. We believe this density imbalance issue significantly contributes to the performance gap observed in these two categories.

---

> > ### Comment · Reviewer_nEx8 · 2025-08-04
> >
> > I appreciate the author response. They have addressed my concerns. I will keep my original rating.

---

> > > ### Author Response · Authors · 2025-08-06
> > >
> > > Thank you for your time to review our work and for your valuable feedback. We are glad that our rebuttal addressed your concerns and truly appreciate your support for our paper.

---

### Official Review · Reviewer_rq8v · 2025-07-04

**Clarity:** 3
**Significance:** 3
**Originality:** 3
**Rating:** 4
**Confidence:** 3

**Summary:**

This work proposes a method for efficiently and accurately reconstructing 3D point clouds from various types of measurements, such as RGB images and depth maps, by combining a pretrained diffusion prior with Forward Curvature-Matching (FCM)-based adaptive likelihood updates. The approach begins by generating an initial 3D point cloud using DDIM sampling. It then uses a differentiable renderer to compute gradients and curvature of the observation loss, and determines a dynamic step size in a Barzilai–Borwein style, updating the likelihood accordingly. This allows for faster and more stable convergence compared to previous methods that rely on fixed step sizes, and the method can be applied to RGB, depth, and multi-view inputs without retraining the prior model. Experimental results on ShapeNet and CO3D demonstrate superior performance over prior methods.

**Questions:**

See Weaknesses

**Ethical Concerns:**

["NO or VERY MINOR ethics concerns only"]

**Final Justification:**

My concerns have been sufficiently addressed. Therefore, I maintain my original positive rating.

**Limitations:**

See Weaknesses

**Quality:**

3

**Strengths And Weaknesses:**

Strenghts
1) The paper is well-written and easy to follow.
2) The use of FCM along with Barzilai–Borwein-style dynamic step size estimation, combined with a one-time Armijo condition check, is both novel and practical. This overcomes the limitations of fixed step sizes in previous DPS methods while allowing adaptation to various modalities (RGB, depth, multi-view) without retraining the prior model.
3) The modularity of the method—only requiring the renderer to be changed for different modalities—suggests it could be highly useful in practice.

Weaknesses
1) While the use of a differentiable renderer reduces the need for retraining the prior model, this also means that the overall reconstruction quality heavily depends on the renderer's performance. It is unclear how well the method performs under varying lighting conditions across views in multi-view input scenarios.
2) On the real-world CO3D dataset, the evaluation is limited to only two categories: 'hydrant' and 'teddy bear'. A broader evaluation across more categories would strengthen the claims.
3) The proposed method involves several hyperparameters (e.g., $\eta_{FCM}$, $L_\text{est}$, $T$, $\delta_{0}$), but no detailed analysis is provided. Since different hyperparameters are used for different datasets (e.g., $T=256$, $\delta_{0}=2×10^{-2}$ for ShapeNet; $T=512$, $\delta_{0}=6×10^{-3}$ for CO3D), a justification or ablation study is needed to explain how these values were chosen and how sensitive the method is to these choices.

---

> ### Author Rebuttal · Authors · 2025-07-28
>
> Thank you for the detailed review and constructive feedback. The following is our response to the specific comments and questions.
>
> > While the use of a differentiable renderer reduces the need for retraining the prior model, this also means that the overall reconstruction quality heavily depends on the renderer's performance. It is unclear how well the method performs under varying lighting conditions across views in multi-view input scenarios.
>
> You correctly identify that our reconstruction quality depends on the differentiable renderer. This is indeed a deliberate design choice—by decoupling the learned prior from the rendering-based likelihood, we gain modularity and flexibility. As differentiable rendering improves, our method can immediately benefit without retraining. This contrasts with methods like PC$^2$ and BDM that embed rendering assumptions into their learned posteriors.
>
> Regarding lighting variations in multi-view scenarios: Our renderer uses simple alpha compositing (Eq.~6) without complex shading models, making it relatively robust to lighting changes. The point colors we optimize represent diffuse appearance rather than view-dependent effects. Additionally, our multi-view loss (Eq. 12) averages gradients across all views, which naturally balances variations between views.
>
> That said, we acknowledge we haven't explicitly tested extreme lighting variations. Our CO3D experiments include real-world captures with natural lighting differences between views, and the method performs well in these scenarios. However, systematic evaluation under controlled lighting changes would be valuable future work.
>
> For applications requiring explicit lighting modeling, our framework could incorporate more sophisticated renderers that separate geometry and appearance—the modularity of our approach makes such extensions straightforward.
>
> > On the real-world CO3D dataset, the evaluation is limited to only two categories: 'hydrant' and 'teddy bear'. A broader evaluation across more categories would strengthen the claims.
>
> We selected these categories to cover diverse real-world conditions and object characteristics. Hydrant represents an outdoor object with strong geometric symmetry, while teddy bear represents an indoor object with complex, asymmetric geometry. Together, they capture two contrasting settings that we believe are sufficient to demonstrate the robustness and versatility of our method.
>
> That said, we agree that expanding the evaluation to a broader set of categories would further strengthen our claims. We made a sincere attempt to include additional categories during the rebuttal period. However, the computational cost of additional training is substantial, and unfortunately, it could not be accommodated within the limited time available for the rebuttal period.
>
> We will address this in the camera-ready version by conducting an expanded evaluation across more CO3D categories, thereby providing a more comprehensive validation of our method.
>
> > The proposed method involves several hyperparameters (e.g., $\eta_{\text{FCM}}, L_{\text{est}}, T, \delta_0$), but no detailed analysis is provided. Since different hyperparameters are used for different datasets (e.g., $T=256,\delta_0=2\times 10^{-2}$ for ShapeNet; $T=512,\delta_0 = 6\times 10^{-3}$), a justification or ablation study is needed to explain how these values were chosen and how sensitive the method is to these choices.
>
> |$L_{\text{est}}$ | EMD($\times10$) | CD($\times10$) | F-score | $\\eta_{\text{FCM}}$ | EMD($\times10$) | CD($\times10$) | F-score |
> |:---|---|---|---|:---|---|---|---|
> |$L_{\text{est}}=100$ | 0.731 | 0.770 | 0.307 | $\\eta_{\text{FCM}} = 10^{-6}$ | 0.585 | 0.563 | 0.373 |
> |$L_{\text{est}}=10$ | 0.585 | 0.563 | 0.373 | $\\eta_{\text{FCM}} = 10^{-5}$ | 0.579 | 0.566 | 0.377 |
> |$L_{\text{est}}=1$ | 0.588 | 0.564 | 0.376 | $\\eta_{\text{FCM}} = 10^{-4}$ | 0.587 | 0.561 | 0.382 |
> |$L_{\text{est}}=2/3$ | 0.587 | 0.561 | 0.382 | $\\eta_{\text{FCM}} = 10^{-3}$ | 0.586 | 0.565 | 0.370 |
>
> | $\delta_0$ | EMD ($\times10$) | CD ($\times10$) | F-score |
> |:---|---|---|---|
> | $\delta_0=5\times10^{-2}$  | 0.644| 0.615 | 0.343 |
> | $\delta_0=2\times10^{-2}$  | 0.587 | 0.561 | 0.382 |
> | $\delta_0=10^{-2}$ | 0.594 | 0.574 | 0.369 |
> | $\delta_0=10^{-3}$ | 0.665 | 0.660 | 0.330 |
>
> We used different hyperparameters for each dataset based on their characteristics: The CO3D dataset is derived from real-world captures and exhibits complex geometry. Accurately representing such complexity and recovering fine-grained details require a higher point density. In contrast, the ShapeNet dataset consists of synthetic objects with simpler geometry, allowing reconstructions to be achieved with a lower number of points. Given that denser point clouds inherently require more frequent and finer updates during sampling, we selected a smaller ($\delta_0$) and increased the total number of sampling steps ($T$) to achieve optimal reconstruction performance under this experimental configuration.
>
> We provide detailed ablation studies for key hyperparameters ($L_{\text{est}}, \eta_{\text{FCM}}, \delta_0$) in the above tables, conducted on single-view ShapeNet reconstruction. These show our method is robust to moderate parameter variations while indicating preferable configurations. We will ensure these justifications are clearly described in the camera-ready version.

---

> > ### Comment · Reviewer_rq8v · 2025-08-05
> > **Official Comment by Reviewer rq8v**
> >
> > Thank you for the authors’ detailed response. I hope that the hyperparameter analysis experiments conducted during the rebuttal, along with the broader experiments that could not be performed due to time constraints, will be incorporated into the final version.

---

> ### Author Response · Authors · 2025-08-06
>
> Thank you for your valuable feedback and for taking the time to review our work. We greatly appreciate your suggestions regarding the hyperparameter analysis and additional experiments. We will make sure to incorporate these experiments into the final version of our paper to enhance its completeness and rigor. We are committed to addressing your points thoroughly and appreciate your strong support for our work.

---

### Note · Authors · 2025-08-12

We sincerely thank all reviewers for their constructive feedback and positive evaluations. Our paper introduces Forward Curvature-Matching (FCM)—an adjoint-free, scale-adaptive likelihood update inside DDIM—for diffusion-prior 3D reconstruction, improving stability and efficiency. During the rebuttal and discussion phases, we conducted additional experiments, including detailed runtime breakdowns and ablation studies on key hyperparameters (e.g., $L_\text{est}, \delta_0, \eta_\text{FCM}$), demonstrating improved reconstruction accuracy at matched NFEs, competitive end-to-end speed, and a clear quality–time trade-off. Furthermore, the proposed FCM framework is representation-agnostic, making it applicable beyond point clouds.

If accepted, we will incorporate the following improvements into the camera-ready version:

**Expanded CO3D evaluation** (reviewer *rq8v*)
> Broaden beyond two categories to better demonstrate robustness across object types.

**Ablation studies on hyperparameters** (reviewer *rq8v, zdEn, QN2v*)
> Provide sensitivity analyses for $L_\text{est}, \delta_0, \eta_\text{FCM}$, and $k$; explain selection criteria; and report recommended defaults.

**Additional runtime & cost comparison** (reviewer *zdEn*)
> Add per-step and total inference comparisons against baselines, including per-step computation, total inference time, and trade-offs in $k$ between reconstruction quality and speed.

**Clarification of alpha computation** (reviewer *zdEn*)
> Clarify our spatial proximity–based alpha assignment for point clouds and its integration into Equation (6).

**Enhanced visualization of the diffusion process** (reviewer *QN2v*)
> Provide finer trajectory snapshots every 16 or 32 steps (instead of 64) to better illustrate progression.

**Code release**
> Release the full implementation, including training scripts, inference pipelines, and pre-trained models, as open-source on GitHub.

We believe these clarifications and additions further demonstrate that FCM is a practical, theoretically grounded update that improves accuracy, stability, and speed for diffusion-prior 3D reconstruction. We appreciate the reviewers’ engagement and look forward to presenting the strengthened version.

---

### Decision · Program_Chairs · 2025-09-17

**Decision:**

Accept (spotlight)

**Comment:**

The paper presents Forward Curvature-Matching (FCM) that can determine optimal step size dynamically in the diffusion-based 3D point cloud reconstruction problem from single images. The novelty and efficacy of the proposed method are acknowledged by all reviewers. All reviewers recommend acceptance, explicitly noting that all concerns have been addressed during the rebuttal and author-reviewer discussion phase.

Given this strong consensus of reviewers and the clear strengths of the paper, the AC recommends accepting this paper.